# Masking the Gaps: An Imputation-Free Approach to Time Series Modeling with Missing Data

## Abstract

Modeling time series is important in a variety of domains, yet it is challenged by the presence of missing values in real-world time-series datasets. Traditional frameworks for modeling time-series with missing values typically involve a two-step process, where the missing values are first filled-in using some imputation technique, followed by a time-series modeling approach on the imputed time-series. However, existing two-stage approaches suffer from two major drawbacks: first, the propagation of imputation errors into subsequent time-series modeling performance, and second, the inherent trade-offs between imputation efficacy and imputation complexity. To this end, we propose a novel imputation-free approach for handling missing values in time series termed **Miss**ing Feature-aware **T**ime **S**eries **M**odeling (**MissTSM**) with two main innovations. First, we develop a novel embedding scheme that treats every combination of time-step and feature (or channel) as a distinct token, encoding them into a high-dimensional space. Second, we introduce a novel Missing Feature-Aware Attention (MFAA) Layer to learn latent representations at every time-step based on partially observed features. We evaluate the effectiveness of MissTSM in handling missing values over multiple benchmark datasets using two synthetic masking techniques: missing completely at random (MCAR) and periodic masking, and a real-world missing-value dataset.

## 1 Introduction

Multivariate time-series modeling for tasks such as forecasting (Lim & Zohren, 2021; Torres et al., 2021) and classification (Ismail Fawaz et al., 2019) are important in a number of applications including electric load forecasting (Din & Marnerides, 2017; L'Heureux et al., 2022), traffic flow prediction (Xu et al., 2020; Jiang et al., 2023), weather forecasting (Bojesomo et al., 2021), healthcare monitoring (Tonekaboni et al., 2021), and financial analysis (Henrique et al., 2019). There is a growing body of literature on time-series modeling using deep learning approaches, including early adoption of recurrent architectures such as Recurrent Neural Networks (RNNs) (Hewamalage et al., 2021) and Long Short-Term Memory networks (LSTMs) (Hochreiter & Schmidhuber, 1997; Siami-Namini et al., 2019). With the success of transformer-based models (Vaswani et al., 2017) in the domain of natural language modeling, there is a growing trend to use transformers in the domain of time-series (Wen et al., 2022). This includes recent frameworks such as SimMTM (Dong et al., 2024), PatchTST (Nie et al., 2022a), and iTransformers (Liu et al., 2023), which already show superior performance compared to non-transformer baselines on various time-series modeling benchmarks by leveraging self-attention mechanisms for capturing temporal dependencies. However, existing deep learning based methods for time-series modeling typically assume the availability of complete time-series data with no missing values. While such datasets are available for benchmarking experiments, in real-world applications, it is common to observe missing values in the time-series due to several reasons such as sensor malfunctions, communication disruptions, or the prohibitive costs of high-frequency data acquisition across all features. The presence of missing values on arbitrary sets of features at varying time-steps introduces "gaps" in the data that can impair the application of state-of-the-art models unless specific adaptations are made.

A common approach for handling missing values in time-series data is to use imputation methodologies (Ahn et al., 2022; Batista et al., 2002), which are aimed at reconstructing (or filling) the missing

values based on observed data. This includes two broad classes of imputation techniques: those that leverage cross-channel correlations (Batista et al., 2002; Acuna & Rodriguez, 2004) and those that exploit temporal dynamics (Box et al., 2015). Despite their promise, these methods are prone to producing inferior imputations in applications involving complex, nonlinear temporal dynamics and/or intricate inter-dependencies across channels. Recently, deep learning-based approaches for imputation have been developed (Tashiro et al., 2021; Cini et al., 2021; Liu et al., 2019), which can jointly learn the temporal dynamics with cross-channel correlations. These methods rely on a single entangled representation (or hidden state) to model nonlinear dynamics (Woo et al., 2022) which can be a limitation in capturing the multifaceted nature of time-series. Matrix factorization based techniques (Liu et al., 2022) have also been proposed that offer disentangled temporal representations, enhancing the ability to differentiate and model distinct temporal features such as trends, seasonality, and local bias. However, all existing frameworks for time-series modeling with missing values are inherently two stage approaches, where the missing values of the time-series are first filled-in using an imputation technique, followed by feeding the imputed time-series to a time-series modeling approach (see Figure 1). This two-stage approach introduces two critical challenges: *first*, the propagation of imputation errors into subsequent time-series modeling performance, and *second*, the inherent trade-offs between imputation efficacy and imputation complexity.

To overcome the inherent limitations of imputation-based techniques, we ask the question: *"can we circumvent the need for imputation by designing a deep learning framework that can directly model time-series with missing values?"* To answer this question, we draw inspiration from the recent success of masked modeling approaches in domains including vision (He et al., 2022) and language (Devlin et al., 2018) where *"masked-attention"* operations embedded in Transformer blocks are effectively utilized to reconstruct data from partial observations. Based on this insight, we propose a novel **Miss**ing Feature-aware **T**ime **S**eries **M**odeling (**MissTSM**) Framework, which capitalizes on the information contained in partially observed features to perform downstream time-series modeling

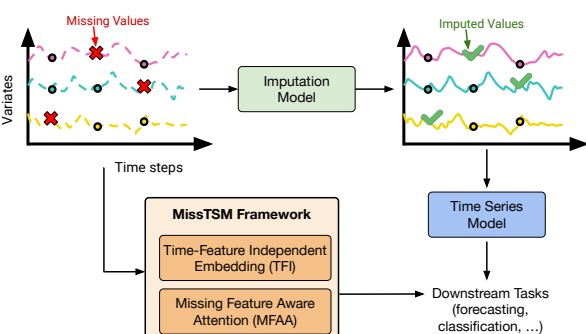

Figure 1: Comparing the traditional two-stage approach of time-series modeling with missing values with our single-stage MissTSM framework that does not require imputations.

tasks without explicitly imputing the missing values (see Figure 1). MissTSM performs end-to-end modeling of time-series with missing values using two main innovations. *First*, we develop a novel embedding scheme, termed *Time-Feature Independent (TFI) Embedding*, which treats every combination of time-step and feature (or channel) as a distinct token, encoding them into a high-dimensional space. *Second*, we introduce a novel *Missing Feature-Aware Attention (MFAA) Layer* to learn latent representations at every time-step based on partially observed features. Additionally, we use the framework of Masked Auto-encoder (MAE) (He et al., 2022) to perform self-supervised learning of latent representations for time-series reconstruction, which can be re-used for downstream tasks such as forecasting and classification. To evaluate the ability of MissTSM to model time-series with missing values, we consider two synthetic masking techniques: missing completely at random (MCAR), and periodic masking, to simulate varying scenarios of missing values. We show that MissTSM achieves comparable performance as state-of-the-art models on multiple benchmark datasets and on a real-world dataset with a high degree of missing values without using any imputation techniques.

## 2 RELATED WORKS

**Imputation Methods:** Time-series modeling, be it for classification or forecasting, has several applications in a number of domains. Traditionally, most imputation techniques for handling missing values in time-series are based on statistical approaches involving either single-value imputation, such as mean or median filling (Fung, 2006), proximity-based imputation (Batista et al., 2002), or multiple-value imputation, such as Maximum Likelihood Imputation (Dempster et al., 1977) and Matrix completion methods (Mnih & Salakhutdinov, 2007). In recent years, there is a growing trend

to use machine learning techniques for time-series imputations, e.g., Bayesian Networks (Sardinha et al., 2018), Random Forest (Stekhoven & Bühlmann, 2012), and deep learning methods such as SAITS (Du et al., 2023), GAIN(Yoon et al., 2018a), and BRITS(Cao et al., 2018). While these deep learning-based models are highly efficient during inference, they require additional training time, which add to the already large time complexity of time-series models.

**Time-series Modeling:** While there are several time-series modeling approaches for short-term forecasting (Challu et al., 2023) (Oreshkin et al., 2019), there is a growing focus to explore long-term forecasting problems. With the introduction of attention mechanisms via transformer models (Vaswani et al., 2017), a number of transformer-based time-series models have been introduced in the last few years (Wu et al., 2021), (Nie et al., 2022b), (Dong et al., 2024), (Liu et al., 2023) for longer-horizon forecasting. The current state-of-the-art methods can be divided into three primary categories— Representation Learning methods, Transformer-based methods, and LLM-based methods, with the latter being the newest addition to the group. Representation Learning methods, especially MAE-style time-series models (Li et al., 2023), are receiving a lot of recent interest owing to their ability of learning both low-level and high-level representations useful for varied downstream tasks such as forecasting and classification. However, a major limitation restricting their applicability to real-world problems is the presence of missing values common to many time-series datasets. While state-of-the art time-series models provide strong prediction accuracy, they are not designed to handle missing values. Transformer-based models, including the ones that use self-supervised style training, can deal with missing values in the temporal domain, but not in the feature-space. There have also been models like (Liu et al., 2023) that efficiently captures the feature-space correlations and show superior forecasting results. However, they do not account for the presence of missing values along the temporal dimension of the time-series variates. In contrast to all existing approaches, our work accounts for missing values in both the feature and temporal space through the proposed Time-Feature Independent Embedding and Missing Feature-Aware Attention Mechanisms.

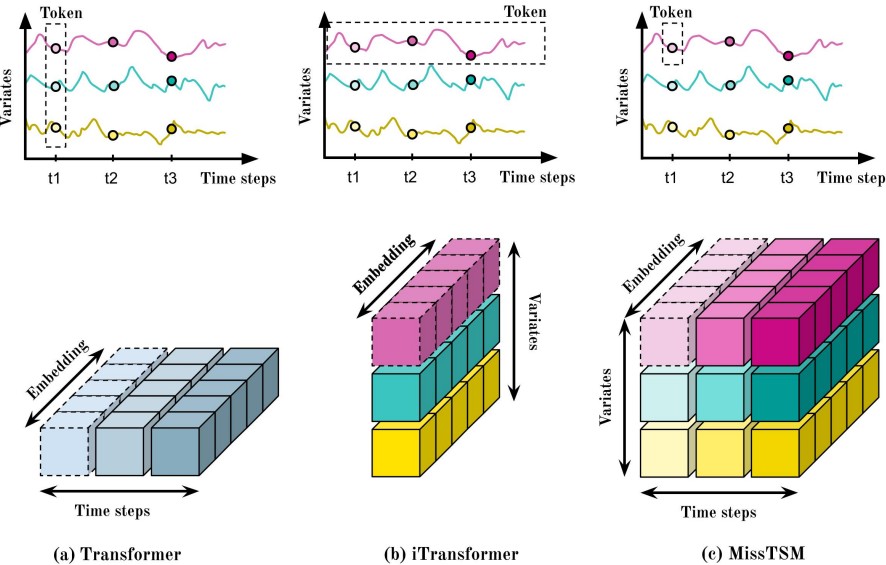

Figure 2: Schematic of the Time-Feature Independent (TFI) Embedding of MissTSM that learns a different embedding for every combination of time-step and variate, in contrast to the time-only embeddings of Transformer (Vaswani et al., 2017) and the variate-only embeddings of iTransformers (Liu et al., 2023).

## 3 MISSING FEATURE TIME-SERIES MODELING (MISSTSM)

### 3.1 NOTATIONS AND PROBLEM FORMULATIONS

Let us represent a multivariate time-series as $\mathbf{X} \in \mathbb{R}^{T \times N}$, where $T$ is the number of time-steps, and $N$ is the dimensionality (number of variates) of the time-series. We assume a subset of variates (or

features) to be missing at some time-steps of $\mathbf{X}$, represented in the form of a missing-value mask $\mathcal{M} \in [0,1]^{T \times N}$, where $\mathcal{M}_{(t,d)}$ represents the value of the mask at $t$-th time-step and $d$-th dimension. $\mathcal{M}_{(t,d)} = 1$ denotes that the corresponding value in $\mathbf{X}_{(t,d)}$ is missing, while $\mathcal{M}_{(t,d)} = 0$ denotes that $\mathbf{X}_{(t,d)}$ is observed. Furthermore, let us denote $\mathbf{X}_{(t,:)} \in \mathbb{R}^N$ as the multiple variates of the time-series at a particular time-step $t$, and $\mathbf{X}_{(:,d)} \in \mathbb{R}^T$ as the uni-variate time-series for the variate $d$. In this paper, we consider two downstream tasks for time-series modeling: forecasting and classification. For forecasting, the goal is to predict the future $S$ time-steps of $\mathbf{X}$ represented as $\mathbf{Y} \in \mathbb{R}^{S \times N}$. Alternatively, for time-series classification, the goal is to predict output labels $\mathbf{Y} \in \{1, 2, ..., C\}$ given $\mathbf{X}$, where $C$ is the number of classes.

## 3.2 Learning Embeddings for Time-Series with Missing Features

**Limitations of Existing Methods:** The first step in time-series modeling using transformer-based architectures is to learn an embedding of the time-series $\mathbf{X}$ that can be sent to the transformer encoder. Traditionally, this is done using an Embedding-layer (typically implemented using a multi-layered perceptron) as $\texttt{Embedding} : \mathbb{R}^N \mapsto \mathbb{R}^D$ that maps $\mathbf{X} \in \mathbb{R}^{T \times N}$ to the embedding $\mathbf{H} \in \mathbb{R}^{T \times D}$, where $D$ is the embedding dimension. The Embedding layer operates on every time-step independently such that the set of variates observed at time-step $t$, $\mathbf{X}_{(t,:)}$, is considered as a single token and mapped to the embedding vector $\mathbf{h}_t \in \mathbb{R}^D$ as $\mathbf{h}_t = \texttt{Embedding}(\mathbf{X}_{(t,:)})$ (see Figure 2(a)). An alternate embedding scheme was recently introduced in the framework of inverted Transformer (iTransformer) (Liu et al., 2023), where the uni-variate time-series for the $d$-th variate, $\mathbf{X}_{(:,d)}$, is considered as a single token and mapped to the embedding vector: $\mathbf{h}_d = \texttt{Embedding}(\mathbf{X}_{(:,d)})$ (see Figure 2(b)). While both these embedding schemes have their unique advantages, they are unfit to handle time-series with arbitrary sets of missing values at every time-step. In particular, the input tokens to the Embedding layer of Transformer or iTransformer requires all components of $\mathbf{X}_{(t,:)}$ or $\mathbf{X}_{(:,d)}$ to be observed, respectively. If any of the components in these tokens are missing, we will not be able to compute their embeddings and thus will have to discard either the time-step or the variate, leading to loss of information.

**Time-Feature Independent (TFI) Embedding:** To address this challenge, we propose a novel *Time-Feature Independent (TFI) Embedding* scheme for time-series with missing features, where the value at each combination of time-step $t$ and variate $d$ is considered as a single token $\mathbf{X}_{(t,d)}$, and is independently mapped to an embedding using $\texttt{TFIEmbedding} : \mathbb{R} \mapsto \mathbb{R}^D$ as follows:

$$\mathbf{h}_{(t,d)} = \texttt{TFIEmbedding}(\mathbf{X}_{(t,d)}) \tag{1}$$

In other words, the $\texttt{TFIEmbedding}$ Layer maps $\mathbf{X} \in \mathbb{R}^{T \times N}$ into the TFI embedding $\mathbf{H}^{\text{TFI}} \in \mathbb{R}^{T \times N \times D}$ (see Figure 2(c)). We should emphasize that we apply the $\texttt{TFIEmbedding}$ Layer only on tokens $\mathbf{X}_{(t,d)}$ that are observed or not missing (i.e., $\mathcal{M}_{(t,d)} = 0$). The advantage of such an approach is that even if a particular value in the time-series is missing, other observed values in the time-series can be embedded *"independently"* without being affected by the missing values. Later, we demonstrate how our proposed Missing Feature-Aware Attention (MFAA) Layer takes advantage of the TFI embedding scheme to compute masked cross-attention among the observed features at a time-step to account for the missing features.

**2D Positional Encodings:** We add Positional Encoding vectors $\mathbf{PE}$ to the TFI embedding $\mathbf{H}^{\text{TFI}}$ to obtain positionally-encoded embeddings, $\mathbf{Z} = \mathbf{PE} + \mathbf{H}^{\text{TFI}}$. Since TFI embeddings treat every time-feature combination as a token, we use a 2D-positional encoding scheme defined as follows:

$$\texttt{PE}(t, d, 2i) = \sin\left(\frac{t}{10000^{(4i/D)}}\right); \quad \texttt{PE}(t, d, 2i+1) = \cos\left(\frac{t}{10000^{(4i/D)}}\right), \tag{2}$$

$$\texttt{PE}(t, d, 2j + D/2) = \sin\left(\frac{d}{10000^{(4j/D)}}\right); \quad \texttt{PE}(t, d, 2j+1+D/2) = \cos\left(\frac{d}{10000^{(4j/D)}}\right), \tag{3}$$

where $t$ is the time-step, $d$ is the feature, and $i, j \in [0, D/4)$ are integers.

## 3.3 Missing Feature-Aware Attention (MFAA) Layer

We propose a novel *Missing Feature-Aware Attention (MFAA) Layer* illustrated in Figure 3 to leverage the power of *"masked-attention"* for learning latent representations at every time-step using partially

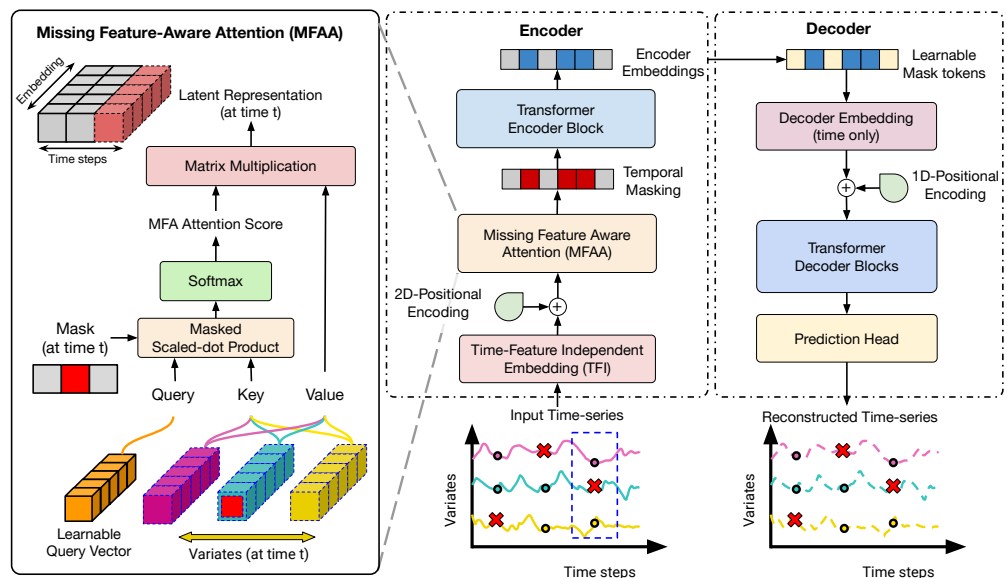

Figure 3: A schematic illustration of the overall MissTSM Framework with a zoomed-in view of the Missing Feature-Aware Attention (MFAA) Layer on the left.

observed features. MFAA works by computing *attention scores* based on the partially observed features at a time-step $t$, which are then used to perform a weighted sum observed features to obtain the latent representation $\mathbf{L}_t$. As shown in Figure 3, these latent representations are later fed into a encoder-decoder based self-supervised learning framework to reconstruct time-series. This is analogous to traditional imputation techniques that utilize cross-channel correlations for imputations. However, the difference in our framework is that once we have learned latent representations $\mathbf{L}_t$ using the paradigm of self-supervised learning, we can directly use them for downstream tasks without explicitly imputing the missing values.

**Mathematical Formulation:** To obtain *attention scores* from partially-observed features at a time-step, we apply a masked scaled-dot product operation followed by a softmax operation described as follows. We first define a learnable query vector $\mathbf{Q} \in \mathbb{R}^{1 \times D}$ which is independent of the variates and time-steps. The positionally-encoded embeddings at time-step $t$, $\mathbf{Z}_{(t,:)}$, are used as key and value inputs in the MFAA Layer. Specifically, The query, key, and value vectors are defined using linear projections as follows:

$$\hat{\mathbf{Q}} = \mathbf{Q}\mathbf{W}^{\mathbf{Q}}, \quad \hat{\mathbf{K}}_t = \mathbf{Z}_{(t,:)}\mathbf{W}^{\mathbf{K}}, \quad \hat{\mathbf{V}}_t = \mathbf{Z}_{(t,:)}\mathbf{W}^{\mathbf{V}}, \tag{4}$$

Here, $\hat{\mathbf{Q}} \in \mathbb{R}^{1 \times d_k}$ and $\hat{\mathbf{K}}_t, \hat{\mathbf{V}}_t \in \mathbb{R}^{N \times d_k}$, where $d_k$ is the dimension of the vectors after linear projection. The linear projection matrices for the query, key, and values are defined as: $\mathbf{W}^{\mathbf{Q}}, \mathbf{W}^{\mathbf{K}}, \mathbf{W}^{\mathbf{V}} \in \mathbb{R}^{D \times d_k}$ respectively. Note that the key $\hat{\mathbf{K}}_t$ and value $\hat{\mathbf{V}}_t$ vectors depend on the time-step $t$, while the query vector doesn't change with time. We then define the Missing Feature-Aware Attention (MFAA) Score at a given time-step $t$ as a masked scalar dot-product of the query and key vector followed by normalization of the scores using a Softmax operation, formally defined as follows:

$$\mathbf{A}_t = \mathtt{MFAAScore}(\hat{\mathbf{Q}}, \hat{\mathbf{K}}_t, \mathcal{M}_{(t,:)}) = \mathtt{Softmax}\left(\frac{\hat{\mathbf{Q}}\hat{\mathbf{K}}_t^{\top}}{\sqrt{d_k}} + \eta\mathcal{M}_{(t,:)}\right), \tag{5}$$

where $\mathbf{A}_t \in \mathbb{R}^N$ is the MFAA Score vector of size $N$ corresponding to the $N$ variates, and $\eta \to -\infty$ is a large negative bias. The negative bias term $\eta$ forces the masked-elements that correspond to the missing variates in the time-series to have an attention score of zero. Thus, by definition, the $i$-th element of the MFAA Score $\mathbf{A}_{(t,i)} \neq 0 \implies \mathcal{M}_{(t,:)} \neq 0$. We compute the latent representation $\mathbf{L}_t$

as a weighted sum of the MFAA score $\mathbf{A}_t$ and the Value vector $\hat{\mathbf{V}}_t$ as follows:

$$\mathbf{L}_t = \text{MFAA}(\mathbf{A}_t, \hat{\mathbf{V}}_t) = \mathbf{A}_t \hat{\mathbf{V}}_t \in \mathbb{R}^{d_k} \tag{6}$$

Similar to multi-head attention used in traditional transformers, we extend MFAA to multiple heads as follows:

$$\texttt{MultiHeadMFAA}(\mathbf{Q}, \mathbf{Z}_{(t,:)}, \mathcal{M}_{(t,:)}) = \text{Concat}(\mathbf{L}_t^0, \mathbf{L}_t^1, ..., \mathbf{L}_t^h)\mathbf{W^O} \tag{7}$$

where $h$ is the number of heads, $\mathbf{W^O} \in \mathbb{R}^{hd_k \times D_o}$, $\mathbf{L}_t^i$ is the latent representation obtained from the $i$-th MHAA Layer, and $D_o$ is the output-dimension of the MultiHeadMFAA Layer.

## 3.4 PUTTING EVERYTHING TOGETHER: OVERALL FRAMEWORK OF MISSTSM

Figure 3 shows the overall framework of MissTSM. For an input time-series $\mathbf{X}$, we apply the TFI embedding layer followed by the MFAA layer to learn a latent representation for every time-step. We then integrate the latent representations into a Masked Auto-Encoder (MAE) (He et al., 2022) framework adapted for time-series (similar to Ti-MAE (Li et al., 2023)). Although the design of the proposed TFI-Embedding and MFAA Layers are flexible enough that they can be integrated with any transformer-based time-series modeling framework (e.g., AutoFormer (Wu et al., 2021)), we opted for a masked time-series modeling approach (such as Ti-MAE (Li et al., 2023) and SimMTM (Dong et al., 2024)) due to their recent success in time-series modeling. Further, out of the several state-of-the-art masked time-series modelling techniques, we intentionally chose the simplest variation of MAE, namely Ti-MAE (Li et al., 2023), to highlight the power of TFI and MFAA layers in handling missing values. To the best of our knowledge, MissTSM is the first end-to-end framework for modeling time-series with missing values without explicitly imputing the time-series. Like a typical masked time-series modeling approach, MissTSM has two main stages: (1) *Self-Supervised Learning Stage:* where the multivariate time-series (with missing values) is reconstructed using an encoder-decoder architecture, with the goal of learning meaningful representations using just the encoder, and (2) *Fine-tuning Stage:* where the latent representations learned by the encoder are fed into a multi-layer perceptron to perform downstream tasks of forecasting and classification.

## 4 EXPERIMENTS

**Datasets:** We consider three popular time-series forecasting datasets: ETTh2, ETTm2 (Zhou et al., 2021) and Weather (wea). We follow a train/val/test split ratio of 6:2:2 for ETT datasets and 7:1:2 for Weather dataset following prior work on Autoformer (Wu et al., 2021). For classification, we use three real-world datasets, namely, Epilepsy (Andrzejak et al., 2001), EMG (Goldberger et al., 2000), and Gesture (Liu et al., 2009), and follow the same evaluation setups as proposed in TF-C (Zhang et al., 2022). We also evaluate on a real-world dataset-PhysioNet-2012 (Silva et al., 2012) that contains 12k multivariate clinical time-series samples with 37 variables. See Appendix A.1 for additional details about dataset and evaluation setup description.

**Baselines:** For all of our experiments, we consider five state-of-the-art time-series modeling baselines, SimMTM (Dong et al., 2024), PatchTST (Nie et al., 2022b), AutoFormer (Wu et al., 2021), DLinear (Zeng et al., 2023), and iTransformer (Liu et al., 2023). In order to apply these methods on data with missing values, we also consider four imputation techniques—a $2^{\text{nd}}$-order spline imputation (McKinley & Levine, 1998), a simple k-Nearest Neighbor approach (Tan et al., 2019), and two state-of-the-art deep learning-based imputation techniques, SAITS (Du et al., 2023) and BRITS (Cao et al., 2018). For forecasting, we used a default lookback window of $L = 336$, while we varied the horizon windows as $T \in \{96, 192, 336, 720\}$.

## 4.1 SYNTHETIC MASKING SCHEMES

To simulate varying scenarios of missing values appearing in real-world time-series datasets, we adopt two synthetic masking schemes that we apply on benchmark datasets, namely missing completely at random (MCAR) masking (Little & Rubin, 1987) and periodic masking, described in the following.

**Missing Completely At Random (MCAR) Masking:** In this scheme, we randomly mask out data from a dataset based on a uniform probability $p$ of observing missing values at any time-feature

Table 1: Comparing forecasting performance of baseline methods using mean squared error (MSE) as the evaluation metric under no masking, MCAR masking, and periodic masking. For every dataset, we consider multiple forecasting horizons, $T \in \{96, 192, 336, 720\}$. Results are color-coded as Best , Second best . We report the mean and standard deviations (in brackets) across 5 random sampling of the masking schemes. Subscript $_{SP}$ refer to Spline and $_{SA}$ refer to SAITS

| | | ETTh2 | | | | ETTm2 | | | | Weather | | | | Avg Rank |
|---|---|---|---|---|---|---|---|---|---|---|---|---|---|---|
| | | 96 | 192 | 336 | 720 | 96 | 192 | 336 | 720 | 96 | 192 | 336 | 720 | |
| No Masking | MissTSM | 0.255 | 0.234 | 0.316 | 0.305 | 0.183 | 0.209 | 0.261 | 0.311 | 0.164 | 0.210 | 0.254 | 0.324 | 1.9 |
| | SimMTM | 0.295 | 0.356 | 0.375 | 0.404 | 0.172 | 0.223 | 0.282 | 0.374 | 0.163 | 0.203 | 0.255 | 0.326 | 2.9 |
| | PatchTST | 0.274 | 0.338 | 0.330 | 0.378 | 0.164 | 0.220 | 0.277 | 0.367 | 0.151 | 0.196 | 0.249 | 0.319 | 1.7 |
| | AutoFormer | 0.501 | 0.516 | 0.565 | 0.462 | 0.352 | 0.337 | 0.494 | 0.474 | 0.306 | 0.434 | 0.437 | 0.414 | 5.9 |
| | DLinear | 0.288 | 0.383 | 0.447 | 0.605 | 0.168 | 0.224 | 0.299 | 0.414 | 0.175 | 0.219 | 0.265 | 0.323 | 4.1 |
| | iTransformer | 0.304 | 0.392 | 0.425 | 0.415 | 0.176 | 0.246 | 0.289 | 0.379 | 0.163 | 0.203 | 0.256 | 0.326 | 4.5 |
| MCAR Masking | MissTSM | $0.243_{0.006}$ | $0.259_{0.002}$ | $0.283_{0.009}$ | $0.329_{0.011}$ | $0.224_{0.005}$ | $0.253_{0.009}$ | $0.293_{0.019}$ | $0.316_{0.014}$ | $0.191_{0.003}$ | $0.234_{0.006}$ | $0.281_{0.004}$ | $0.322_{0.008}$ | 2.7 |
| | SimMTM$_{SP}$ | $0.309_{0.001}$ | $0.372_{0.005}$ | $0.396_{0.01}$ | $0.418_{0.008}$ | $0.185_{0.001}$ | $0.243_{0.002}$ | $0.298_{0.001}$ | $0.388_{0.005}$ | $0.203_{0.009}$ | $0.242_{0.010}$ | $0.284_{0.008}$ | $0.386_{0.008}$ | 5.0 |
| | SimMTM$_{SA}$ | $0.457_{0.06}$ | $0.510_{0.061}$ | $0.503_{0.055}$ | $0.472_{0.066}$ | $0.287_{0.037}$ | $0.320_{0.035}$ | $0.342_{0.017}$ | $0.413_{0.014}$ | $0.187_{0.002}$ | $0.240_{0.001}$ | $0.280_{0.001}$ | $0.385_{0.004}$ | 6.2 |
| | PatchTST$_{SP}$ | $0.290_{0.003}$ | $0.355_{0.003}$ | $0.345_{0.003}$ | $0.390_{0.003}$ | $0.169_{0.001}$ | $0.228_{0.001}$ | $0.286_{0.001}$ | $0.378_{0.001}$ | $0.183_{0.009}$ | $0.226_{0.009}$ | $0.277_{0.009}$ | $0.339_{0.004}$ | 2.1 |
| | PatchTST$_{SA}$ | $0.440_{0.059}$ | $0.484_{0.057}$ | $0.434_{0.059}$ | $0.436_{0.075}$ | $0.324_{0.05}$ | $0.362_{0.045}$ | $0.410_{0.049}$ | $0.462_{0.047}$ | $0.175_{0.002}$ | $0.211_{0.000}$ | $0.264_{0.002}$ | $0.335_{0.001}$ | 4.6 |
| | AutoFormer$_{SP}$ | $0.559_{0.05}$ | $0.628_{0.101}$ | $0.525_{0.037}$ | $0.550_{0.143}$ | $0.280_{0.006}$ | $0.390_{0.158}$ | $0.360_{0.018}$ | $0.475_{0.033}$ | $0.321_{0.008}$ | $0.413_{0.013}$ | $0.508_{0.036}$ | $0.467_{0.032}$ | 8.9 |
| | AutoFormer$_{SA}$ | $0.767_{0.126}$ | $0.526_{0.06}$ | $0.550_{0.019}$ | $0.449_{0.010}$ | $0.610_{0.312}$ | $0.850_{0.365}$ | $0.615_{0.151}$ | $1.045_{0.262}$ | $0.353_{0.013}$ | $0.413_{0.006}$ | $0.474_{0.028}$ | $0.504_{0.049}$ | 10.2 |
| | DLinear$_{SP}$ | $0.296_{0.003}$ | $0.401_{0.018}$ | $0.445_{0.006}$ | $0.607_{0.013}$ | $0.458_{0.169}$ | $0.228_{0.001}$ | $0.302_{0.000}$ | $0.531_{0.144}$ | $0.205_{0.007}$ | $0.241_{0.007}$ | $0.282_{0.008}$ | $0.373_{0.009}$ | 6.5 |
| | DLinear$_{SA}$ | $0.454_{0.053}$ | $0.514_{0.053}$ | $0.542_{0.064}$ | $0.680_{0.084}$ | $0.330_{0.065}$ | $0.365_{0.062}$ | $0.427_{0.058}$ | $0.538_{0.063}$ | $0.190_{0.001}$ | $0.233_{0.000}$ | $0.276_{0.000}$ | $0.333_{0.001}$ | 6.8 |
| | iTransformer$_{SP}$ | $0.313_{0.004}$ | $0.394_{0.014}$ | $0.436_{0.005}$ | $0.429_{0.005}$ | $0.178_{0.001}$ | $0.243_{0.0004}$ | $0.293_{0.001}$ | $0.384_{0.008}$ | $0.197_{0.006}$ | $0.260_{0.007}$ | $0.315_{0.008}$ | $0.349_{0.006}$ | 4.9 |
| | iTransformer$_{SA}$ | $0.492_{0.058}$ | $0.545_{0.048}$ | $0.579_{0.049}$ | $0.540_{0.094}$ | $0.369_{0.080}$ | $0.432_{0.083}$ | $0.482_{0.063}$ | $0.541_{0.075}$ | $0.191_{0.002}$ | $0.228_{0.002}$ | $0.273_{0.002}$ | $0.348_{0.003}$ | 7.7 |
| Periodic Masking | MissTSM | $0.246_{0.018}$ | $0.263_{0.017}$ | $0.301_{0.042}$ | $0.353_{0.015}$ | $0.227_{0.006}$ | $0.249_{0.006}$ | $0.282_{0.011}$ | $0.337_{0.036}$ | $0.212_{0.007}$ | $0.256_{0.008}$ | $0.313_{0.009}$ | $0.379_{0.019}$ | 4.1 |
| | SimMTM$_{SP}$ | $0.372_{0.122}$ | $0.469_{0.198}$ | $0.496_{0.198}$ | $0.510_{0.200}$ | $0.192_{0.010}$ | $0.247_{0.009}$ | $0.301_{0.008}$ | $0.391_{0.008}$ | $0.182_{0.004}$ | $0.248_{0.006}$ | $0.291_{0.009}$ | $0.344_{0.005}$ | 4.7 |
| | SimMTM$_{SA}$ | $0.591_{0.132}$ | $0.666_{0.152}$ | $0.681_{0.182}$ | $0.667_{0.222}$ | $0.389_{0.071}$ | $0.409_{0.054}$ | $0.436_{0.076}$ | $0.505_{0.055}$ | $0.178_{0.002}$ | $0.214_{0.001}$ | $0.261_{0.001}$ | $0.354_{0.003}$ | 6.0 |
| | PatchTST$_{SP}$ | $0.328_{0.047}$ | $0.389_{0.040}$ | $0.381_{0.050}$ | $0.426_{0.058}$ | $0.174_{0.004}$ | $0.231_{0.003}$ | $0.289_{0.004}$ | $0.381_{0.004}$ | $0.181_{0.004}$ | $0.227_{0.005}$ | $0.267_{0.005}$ | $0.346_{0.003}$ | 2.4 |
| | PatchTST$_{SA}$ | $0.581_{0.120}$ | $0.620_{0.132}$ | $0.592_{0.170}$ | $0.644_{0.230}$ | $0.423_{0.054}$ | $0.457_{0.042}$ | $0.493_{0.037}$ | $0.527_{0.027}$ | $0.171_{0.002}$ | $0.212_{0.001}$ | $0.263_{0.005}$ | $0.334_{0.001}$ | 5.2 |
| | Autoformer$_{SP}$ | $0.482_{0.041}$ | $0.685_{0.165}$ | $0.621_{0.166}$ | $0.546_{0.035}$ | $0.329_{0.109}$ | $0.315_{0.010}$ | $0.398_{0.090}$ | $0.456_{0.021}$ | $0.333_{0.0176}$ | $0.387_{0.035}$ | $0.406_{0.025}$ | $0.453_{0.016}$ | 7.5 |
| | Autoformer$_{SA}$ | $1.415_{0.807}$ | $0.810_{0.269}$ | $1.364_{0.760}$ | $0.820_{0.467}$ | $1.303_{1.278}$ | $0.933_{0.444}$ | $1.788_{0.538}$ | $0.809_{0.431}$ | $0.335_{0.009}$ | $0.387_{0.017}$ | $0.435_{0.035}$ | $0.467_{0.017}$ | 10.8 |
| | DLinear$_{SP}$ | $0.346_{0.069}$ | $0.475_{0.108}$ | $0.477_{0.044}$ | $0.649_{0.068}$ | $0.327_{0.188}$ | $0.230_{0.002}$ | $0.305_{0.003}$ | $0.473_{0.038}$ | $0.215_{0.018}$ | $0.244_{0.013}$ | $0.284_{0.008}$ | $0.339_{0.007}$ | 5.0 |
| | DLinear$_{SA}$ | $0.605_{0.109}$ | $0.674_{0.11}$ | $0.728_{0.138}$ | $0.911_{0.158}$ | $0.447_{0.049}$ | $0.475_{0.043}$ | $0.523_{0.042}$ | $0.626_{0.032}$ | $0.190_{0.001}$ | $0.233_{0.000}$ | $0.276_{0.000}$ | $0.333_{0.001}$ | 7.4 |
| | iTransformer$_{SP}$ | $0.358_{0.070}$ | $0.435_{0.067}$ | $0.488_{0.096}$ | $0.497_{0.119}$ | $0.180_{0.005}$ | $0.245_{0.006}$ | $0.296_{0.007}$ | $0.384_{0.007}$ | $0.197_{0.009}$ | $0.233_{0.006}$ | $0.288_{0.01}$ | $0.351_{0.010}$ | 4.2 |
| | iTransformer$_{SA}$ | $0.691_{0.143}$ | $0.715_{0.140}$ | $0.763_{0.153}$ | $0.773_{0.201}$ | $0.512_{0.055}$ | $0.578_{0.052}$ | $0.662_{0.05}$ | $0.680_{0.029}$ | $0.194_{0.001}$ | $0.229_{0.004}$ | $0.274_{0.002}$ | $0.350_{0.003}$ | 8.2 |

combination. We vary the probability value to generate synthetically masked datasets with different fractions of missing values.

**Periodic Masking:** Since missing values in time-series follow periodic patterns in many real-world applications (e.g., the seasonal cycles in weather and environmental datasets), we introduce a periodic masking scheme described as follows. We use a sine curve to generate the masking periodicity with given phase and frequency values for different features. Specifically, the time-dependent periodic probability of seeing missing values is defined as $\hat{p}(t) = p + \alpha(1 - p)\sin(2\pi\nu t + \phi)$, where, $\phi$ and $\nu$ are randomly chosen across the feature space, $\alpha$ is a scale factor, and $p$ is an offset term. We vary $p$ from low to high values to get different fractions of periodic missing values in the data.

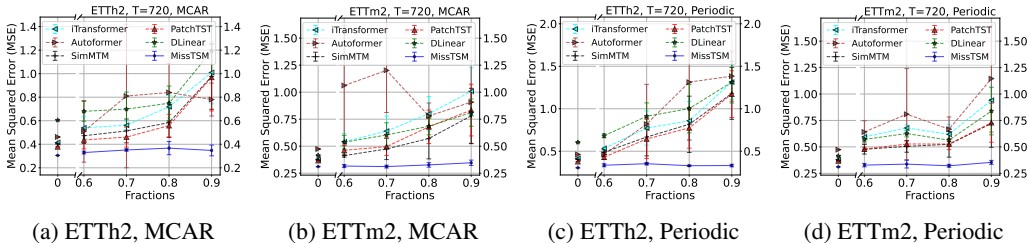

| (a) ETTh2, MCAR | (b) ETTm2, MCAR | (c) ETTh2, Periodic | (d) ETTm2, Periodic |
|---|---|---|---|

Figure 4: **Multiple Time-series Baselines.** Performance comparison between MCAR and Periodic masking with multiple TS Baselines imputed with SAITS. TS Baselines considered: Autoformer Wu et al. (2021), PatchTST Nie et al. (2022b), iTransformer Liu et al. (2023), DLinear Zeng et al. (2023), SimMTM Dong et al. (2024)

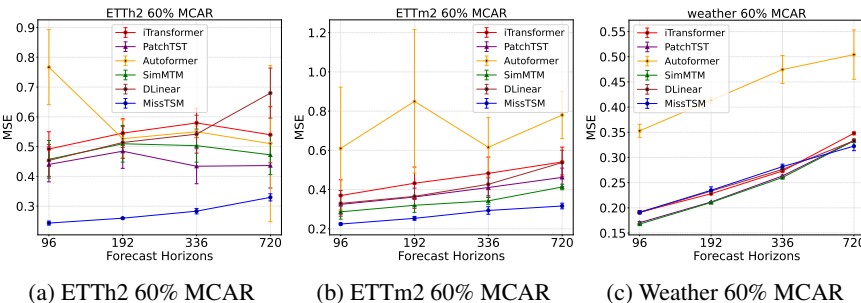

(a) ETTh2 60% MCAR     (b) ETTm2 60% MCAR     (c) Weather 60% MCAR

Figure 5: Forecasting performance with the horizon length $T \in 96, 192, 336, 720$ and fixed lookback length S = 336. Baseline models are imputed with SAITS

## 4.2 RESULTS ON FORECASTING DATASETS

Table 1 compares the forecasting performance of MissTSM with five SOTA baseline methods in terms of the Mean Squared Error (MSE) metric on three datasets (ETTh2, ETTm2 and Weather) with varying forecasting horizons, imputation techniques (Spline and SAITS), and masking schemes. We provide the mean and standard deviations over 5 different samples of the masking schemes. We choose a missing value probability of 60% for MCAR masking and 70% for periodic masking to simulate scenarios with varying (and often extreme) amounts of missing information. We can see that in the no masking experiment, the performance of all methods (with the exception of AutoFormer) are mostly comparable to each other across all three datasets, with MissTSM and PatchTST having a slight edge on the ETTh2/ETTm2 and Weather datasets, respectively. For the MCAR masking experiments, we observe a trend across all the datasets that the MissTSM framework performs slightly better than the baselines for longer-term forecasting (such as forecasting horizon of 720), and comparable to the best-performing baselines on other forecasting horizons. For the Periodic masking experiment, we can see that MissTSM is consistently better than the baselines for ETTh2 dataset, while for the ETTm2 and Weather datasets, the forecasting performance is comparable to the other baselines. These results demonstrate the effectiveness of our proposed MissTSM framework to circumvent the need for explicit imputation of missing values while achieving comparable performance as SOTA.

By being imputation-free, MissTSM does not suffer from the propagation of imputation errors (from the imputation scheme) to forecasting errors (from the time-series models). In Appendix Figure 11, we provide empirical evidence of this error propagation, where we see a positive correlation between imputation errors and forecasting errors of baseline methods, indicating that reducing imputation errors is crucial for improving forecasting accuracy. This finding underscores the limitations of traditional two-stage approaches and suggests that using more sophisticated imputation models is necessary to achieve lower forecasting errors. We also report the computation time of SimMTM (with Spline and SAITS) and MissTSM in Appendix Table 5, where we demonstrate that MissTSM is significantly faster as it does not involve any expensive interpolations as an additional advantage.

## 4.3 ANALYZING THE IMPACT OF MISSING VALUE FRACTIONS ON FORECASTING

To understand the effect of varying masking fractions on the forecasting performance of MissTSM and baseline methods, Figure 4 shows variations in the MSE of five SOTA baseline methods as we increase the missing value fraction in MCAR and periodic masking scheme from 0.6 to 0.9 for forecasting horizons $T = 720$ on two ETT datasets. All of the baselines were trained with imputed time-series using SAITS. We can see that MissTSM mostly performs at par or better than the best-performing time-series baselines (SimMTM), across the two datasets and masking schemes. It is interesting to see that, MissTSM achieves similar MSE as the stronger baseline methods (with more complex architectures both for imputation and time-series modeling) even with extreme 90% missing values.

## 4.4 ANALYZING THE IMPACT OF FORECAST HORIZON

We further analyze the forecasting performance of MissTSM and five SOTA baselines by varying the forecasting horizon $T = \{96, 192, 336, 720\}$ under a 60% MCAR masking scheme, as

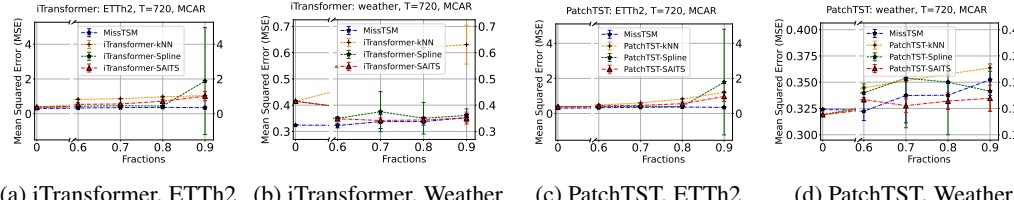

(a) iTransformer, ETTh2  (b) iTransformer, Weather  (c) PatchTST, ETTh2  (d) PatchTST, Weather

Figure 6: **Multiple Imputation Baselines**. Performance comparison across multiple imputation models. Imputation models considered: BRITS Cao et al. (2018), kNN, Spline, SAITS Du et al. (2023). TS Baselines: iTransformer Liu et al. (2023) and PatchTST Nie et al. (2022b)

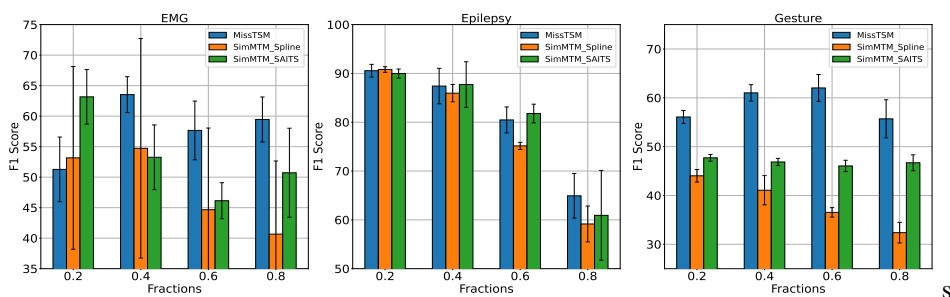

Figure 7: Classification F1 scores on three datasets - EMG, Epilepsy, Gesture. Masking fractions considered: 0.2, 0.4, 0.6, 0.8.

shown in Figure 5. As expected, a common trend across all three datasets is that the forecasting MSE increases with the forecasting horizon for all methods. On the ETTh2 dataset, MissTSM consistently achieves slightly better performance than all other baselines at each forecasting horizon. For the ETTm2 and Weather datasets, MissTSM performs on par with the baselines (except for AutoFormer), further demonstrating that MissTSM can achieve comparable performance to SOTA methods without the need for costly imputation techniques such as SAITS.

### 4.5 COMPARING DIFFERENT IMPUTATION SCHEMES IN TIME-SERIES FORECASTING

The choice of imputation method is critical to overall forecasting performance. While simple imputations like Spline interpolation are computationally inexpensive, they may not provide optimal forecasting accuracy compared to state-of-the-art imputation techniques such as SAITS. In Figure 6, we compare four imputation techniques—Spline, kNN, BRITS, and SAITS—paired with two time-series forecasting models (iTransformer and PatchTST) at a forecasting horizon of $T = 720$. Notably, MissTSM achieves performance comparable to all imputation methods across various masking probabilities on both datasets.

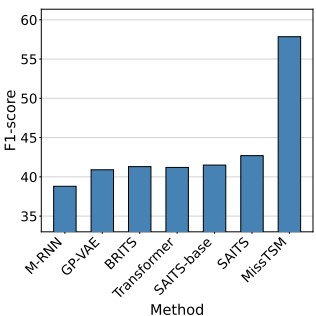

Figure 8: Classification Performance of MissTSM and other imputation baselines on PhysioNet (Silva et al., 2012).

### 4.6 RESULTS ON TIME-SERIES CLASSIFICATION

**Synthetic Benchmarks:** For fine-tuning on the classification tasks, we add a 2-layer multi-layer perceptron to the encoder as the classification layer for our model as well as the SimMTM baseline. As seen from Figure 7, MissTSM achieves roughly similar performance as SimMTM on EMG and Epilepsy datasets, and outperforms SimMTM on the Gesture Dataset.

**Real-world results on Physio-Net:** We compare the performance of the MissTSM framework with six imputation baselines— M-RNN (Yoon et al., 2018b), GP-VAE (Fortuin et al., 2020), BRITS (Cao et al., 2018), Transformer (Vaswani et al., 2017), and SAITS (Du et al., 2023)—on the real-

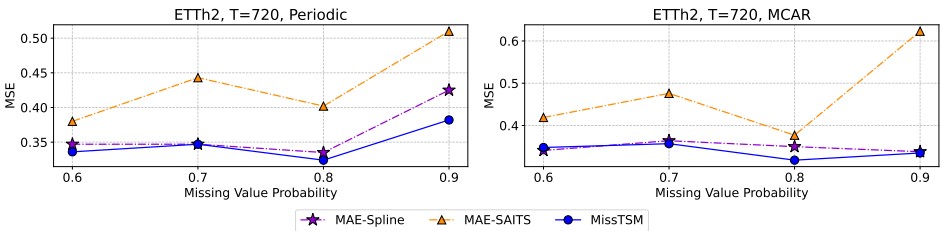

Figure 9: Ablations of MissTSM with and without TFI+MFAA layer on Forecasting datasets.

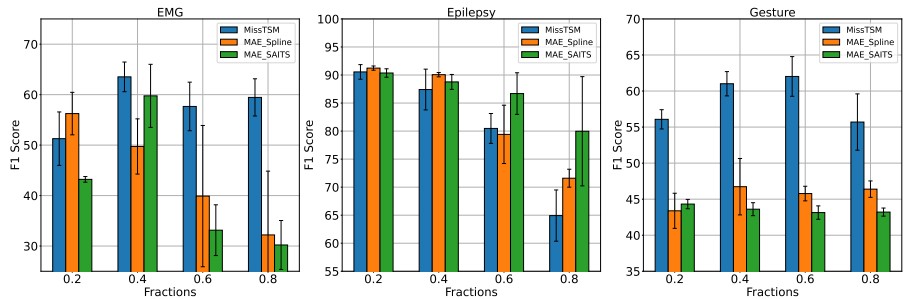

Figure 10: Ablations of MissTSM with and without the TFI+MFAA layer on the classification tasks.

world PhysioNet classification dataset (Silva et al., 2012) that is highly sparse with 80% missing values (see Appendix for additional details), as shown in Figure 8. We follow the same evaluation setup as proposed in (Du et al., 2023). MissTSM achieves an F1-score of 57.84%, representing an approximately 15% improvement over SAITS, the best-performing imputation model, which scored 42.6%. This substantial performance gain on a real-world dataset with missing values highlights the advantages of MissTSM's single-stage approach compared to traditional two-stage methods, beyond synthetic masking schemes used to simulate missing values in other datasets.

### 4.7 ABLATIONS ON FORECASTING & CLASSIFICATION TASKS

In the ablation experiments, our goal is to quantify the effectiveness of the TFI-Embedding scheme and the MFAA Layer on MissTSM. To achieve this, we compare MissTSM with Ti-MAE, which can be viewed as an ablation of MissTSM without the TFI-Embedding and MFAA Layers. We refer to this ablation of MissTSM as MAE. For both the forecasting (see Fig. 9) and classification (see Fig. 10) tasks, we compare the MissTSM framework with MAE trained on spline and SAITS imputation techniques. For forecasting on ETTh2, we observe that our proposed MissTSM framework consistently outperforms the MAE ablations without the MFAA Layer. On the other hand for classification, we show that for all the three datasets, we are either comparable or better than the MAE ablations. This demonstrates the efficacy of the TFI-Embedding and MFAA Layer for time-series modeling with missing values.

## 5 CONCLUSIONS AND FUTURE WORK

To the best of our knowledge, our proposed MissTSM framework is the first end-to-end framework for time-series modeling with missing values that does not require any explicit imputations. We empirically demonstrate the effectiveness of the MissTSM framework across multiple benchmark datasets and synthetic masking strategies, and a real-world dataset with a high degree of missing values. Our proposed framework also has limitations. For example, a limitation of the MFAA layer is that it does not learn the non-linear temporal dynamics and relies on the subsequent transformer encoder blocks to learn the dynamics. Future work can explore modifications of the MFAA layer such that it can jointly learn the cross-channel correlations with the non-linear temporal dynamics.

REPRODUCIBILITY

We have ensured the reproducibility of our work by providing a detailed set of resources. The source code used for training and evaluation, along with instructions for reproducing the experiments, is available via an anonymous GitHub link (https://anonymous.4open.science/r/MissTSM-2-ICLR-64CE/). Comprehensive implementation details, including hyperparameters and compute resources, are described in Appendix. Additionally, we have cited all dataset sources and outlined the data processing steps also in Appendix to facilitate accurate replication of our experiments.

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

# A EXPERIMENTAL SETUP

## A.1 DATASET DESCRIPTION

**Forecasting Dataset Details**

**ETT.** The ETT Zhou et al. (2021) dataset captures load and oil temperature data from electricity transformers. ETTh2 includes 17,420 hourly observations, while ETTm2 comprises 69,680 15-minute observations. Both datasets span two years and contain 7 variates each.

**Weather.** Weather wea is a 10-minute frequency time-series dataset recorded throughout the year 2020 and consists of 21 meteorological indicators, like humidity, air temperature, etc.

Following previous works in this area, we use a train-validation-test split of 6:2:2 for the ETT datasets and 7:1:2 for the Weather dataset. We standardized the input features by subtracting off the mean and dividing by the standard deviation for every feature over the training set. Again, following the approach used in previous works, we compute the MSE in the normalized space of all features considering all features together.

**Classification Dataset Details**

**Epilepsy.** Epilepsy Andrzejak et al. (2001) contains univariate brainwaves (single-channel EEG) sampled from 500 subjects (with 11,500 samples in total), with each sample classified as having epilepsy or not (binary classification).

**Gesture.** Gesture Liu et al. (2009) dataset consists of 560 samples, each having 3 variates (corresponding to the accelerometer data) and each sample corresponding to one of the 8 hand gestures (or classes)

**EMG.** EMG Goldberger et al. (2000) dataset contains 163 EMG (Electromyography) samples corresponding to 3-classes of muscular diseases.

We make use of the following readily available data splits (train, validation, test) for each of the datasets: **Epilepsy** = 60 (30 samples per each class)/20 (10 samples per each class)/11420 (Train/Val/Test) **Gesture** = 320/20/120 (Train/Val/Test) **EMG** = 122/41/41 (Train/Val/Test)

**Physio-Net Dataset:** PhysioNet-2012 Mortality Prediction Challenge (Silva et al., 2012) contains 12k multivariate clinical time-series samples that were collected from patients in ICU. The time-series contains 37 variables, such as temperature, heart rate, blood pressure, etc. that can vary depending on the type of patient. Each of the samples are recorded during the first 48 hours of admission in ICU. Because of the high variability of variables collected across patients in ICU with different patterns of missing information across time, PhysioNet has a high degree of 80% missing values. We follow the experimental setup in (Du et al., 2023), and split the dataset into 80%, 10% and 10% train/val/test split.

## A.2 SYNTHETIC MASKED DATA GENERATION

**Random Masking**: We generated masks by randomly selecting data points across all variates and time-steps, assigning them as missing with a likelihood determined by p (masking fraction). The selected data points were then removed, effectively simulating missing values at random. For multiple runs, we created multiple such versions of the synthetic datasets and compared all baseline methods and MissTSM on the same datasets.

**Periodic Masking**: We use a sine curve to generate the masking periodicity with given phase and frequency values for different features. Specifically, the time-dependent periodic probability of seeing missing values is defined as $\hat{p}(t) = p + \alpha(1 - p)\sin(2\pi\nu t + \phi)$, where, $\phi$ and $\nu$ are randomly chosen across the feature space, $\alpha$ is a scale factor, and p is an offset term. We vary p from low to high values to get different fractions of periodic missing values in the data. To implement this masking strategy, each feature in the dataset was assigned a unique frequency, randomly selected from the range [0.2, 0.8]. This was done to reduce bias and increase randomness in periodicity across the feature space. Additionally, the phase shift was chosen randomly from the range $[0, 2\pi]$. This was applied to each feature to offset the sinusoidal function over time. Like frequency, the phase value

was different for different features. This generated a periodic pattern for the likelihood of missing data.

### A.3 Implementation Details

The experiments have been implemented in PyTorch using NVIDIA TITAN 24 GB GPU. The baselines have been implemented following their official code and configurations. We consider Mean Squared Error (MSE) as the metric for time-series forecasting and F1-score for the classification tasks.

**Forecasting experiments**. MissTSM was trained with the MSE loss, using the Adam Kingma (2014) optimizer with a learning rate of 1e-3 during pre-training for 50 epochs and a learning rate of 1e-4 during finetuning with an early stopping counter of 3 epochs. Batch size was set 16. All the reported missing data experiment results are obtained over 5 trials (5 different masked versions). During fine-tuning for different Prediction lengths (96, 192, 336, 720), we used the same pre-trained encoder and added a linear layer at the top of the encoder.

**Classification experiments**. MissTSM was trained using the Adam Kingma (2014) optimizer, with MSE as the loss function during pre-training and Cross-Entropy loss during fine-tuning. During fine-tuning, we plugged a 64-D linear layer at the top of the pre-trained encoder. We pre-trained and fine-tuned for 100 epochs.

### A.4 Hyper-parameter Details

For MissTSM, we start with the same set of hyper-parameters as reported in the SimMTM paper as initialization (see Table 2), and then search for the best learning rate in factors of 10, and encoder/decoder layers in the range [2, 4]. Note that we only perform hyper-parameter tuning on 100% data, and use the same hyper-parameters for all experiments involving the dataset, such as different missing value probabilities. Our goal is to show the generic effectiveness of our MissTSM framework even without any rigorous hyper-parameter optimization. Additionally, we would also like to note that our model sizes are relatively very small (number of parameters for ETTh2=28,080, Weather= 149,824, and ETTm2= 28,952), compared to other baselines such as SimMTM (ETTh2=4,694,186), iTransformer (ETTh2=254,944), and PatchTST (ETTh2=81,728).

Table 2: Hyperparameters for Forecasting and Classification Tasks

| Task | Enc. Layers | Dec. Layers | Enc. Heads | Dec. Heads | Enc. Embed Dim | Dec. Embed Dim |
|---|---|---|---|---|---|---|
| **Forecasting** | | | | | | |
| ETTh2 | 2 | 2 | 8 | 4 | 8 | 32 |
| ETTm2 | 3 | 2 | 8 | 4 | 8 | 32 |
| Weather | 2 | 2 | 8 | 4 | 64 | 32 |
| **Classification** | | | | | | |
| All Datasets | 3 | 2 | 16 | 16 | 32 | 32 |

## B Additional Results

### B.1 Embedding of 1D data and the Effect of Varying Embedding Sizes

To understand the usefulness of mapping 1D data to multi-dimensional data in TFI embedding, we present (in Table 4) an ablation comparing performances on ETTh2 with and without using high-dimensional projections in TFI Embedding under the no missing value scenario. Projecting 1D scalars independently to higher-dimensional vectors may look wasteful at the time of initialization of TFI Embedding, when the context of time and variates are not incorporated. However, it is during the cross-attention stage (using MFAA layer or later using the Transformer encoder block) that we can leverage the high-dimensional embeddings to store richer representations bringing in the context of time and variate in which every data point resides.

Table 3: Hyper-parameter sensitivity of MissTSM on ETTh2 with 70% Masking Fraction, MCAR. Best results shown in bold, second best underlined. Hyper-parameter settings used in the remainder of experiments in the paper are italicized.

| | Enc. Heads | | | Enc. Layers | | | Enc. Embed Dim | | |
|---|---|---|---|---|---|---|---|---|---|
| | 1 | 4 | 8 | 1 | 2 | 3 | 8 | 16 | 32 |
| 96 | _0.246_ | **0.245** | _0.246_ | 0.249 | ***0.243*** | _0.244_ | ***0.243*** | _0.248_ | 0.285 |
| 192 | **0.261** | 0.273 | _0.266_ | 0.287 | ***0.267*** | _0.271_ | _0.267_ | **0.266** | 0.340 |
| 336 | 0.312 | **0.279** | _0.310_ | **0.294** | _0.392_ | _0.307_ | _0.392_ | **0.316** | _0.369_ |
| 720 | **0.326** | 0.346 | _0.333_ | _0.351_ | ***0.323*** | 0.355 | ***0.323*** | _0.338_ | 0.446 |

| | Dec. Heads | | | Dec. Layers | | | Dec. Embed Dim | | |
|---|---|---|---|---|---|---|---|---|---|
| | 1 | 4 | 8 | 1 | 2 | 3 | 8 | 16 | 32 |
| 96 | 0.261 | ***0.243*** | _0.252_ | 0.276 | ***0.242*** | _0.248_ | _0.250_ | 0.259 | ***0.243*** |
| 192 | 0.276 | ***0.267*** | _0.272_ | **0.266** | _0.268_ | 0.268 | **0.257** | 0.272 | _0.267_ |
| 336 | _0.319_ | _0.392_ | **0.301** | **0.262** | _0.352_ | _0.271_ | _0.289_ | **0.266** | _0.392_ |
| 720 | _0.324_ | ***0.323*** | 0.330 | **0.323** | _0.364_ | _0.341_ | _0.353_ | 0.384 | ***0.323*** |

From Table 4, we can see that TFI embedding with 8-dimensional vectors consistently outperform the ablation with 1D representations, empirically demonstrating the importance of high-dimensional projections in our proposed framework.

Table 4: Effect of TFI Embedding with embedding size=1 and embedding size=8 under no masking scenario. Dataset=ETTh2

| Time Horizon | TFI Embedding with embedding size = 1 | TFI Embedding with embedding size = 8 |
|---|---|---|
| 96 | 0.283 ± 0.048 | **0.245 ± 0.011** |
| 192 | 0.285 ± 0.078 | **0.260 ± 0.023** |
| 336 | 0.319 ± 0.023 | **0.300 ± 0.016** |
| 720 | 0.378 ± 0.022 | **0.334 ± 0.032** |

## C  COMPUTATIONAL COMPLEXITY AND ERROR PROPAGATION

We consider a case study of a classification task on the Epilepsy dataset. Dataset is 80% masked under MCAR. Spline and SAITS are the imputation techniques and SimMTM is the time-series model used. We report the total modeling time as the sum of imputation time and the time-series model training time. The experiments are conducted on NVIDIA TITAN 24 GB GPUs.

In Table 5, we observe that, while SimMTM integrated with SAITS achieves a higher F1 score than Spline, but the total imputation time for SAITS is significantly higher than that of Spline. This additional computational overhead substantially increases the overall modeling time. Moreover, SAITS has approximately 1.3 million trainable parameters, further increasing the overall model complexity of the time-series modeling task. This highlights the potential trade-off between imputation efficiency and complexity (by imputation complexity we are referring to both model and time complexity).

In the case of our proposed method, we do not have the extra overhead of imputation complexity. Simultaneously, MissTSM also achieves superior performance.

Figure 11 captures the propagation of imputation errors and forecasting errors for Weather dataset at 720 forecast horizon. It demonstrates that there is an overall positive correlation between the imputation error and forecasting errors, thereby demonstrating propagation of the imputation errors into the downstream time-series models.

Table 5: Comparison of total computational cost between MissTSM and SimMTM integrated with Spline and SAITS

| Time-Series Model | Imp. Model | Imp. Time (sec) | TS Model Train Time (sec) | Total Time (sec) | F1 Score |
|---|---|---|---|---|---|
| **SimMTM** | SAITS | 949 ± 42.9 | 397.59 ± 2.64 | 1346.59 ± 45.54 | 61.0 ± 9.20 |
| | Spline | 8.74 ± 0.38 | 397.59 ± 2.64 | 406.33 ± 3.02 | 59.16 ± 3.67 |
| **MissTSM** | N/A | N/A | 346.8 ± 7.32 | **346.8 ± 7.32** | **64.93 ± 4.57** |

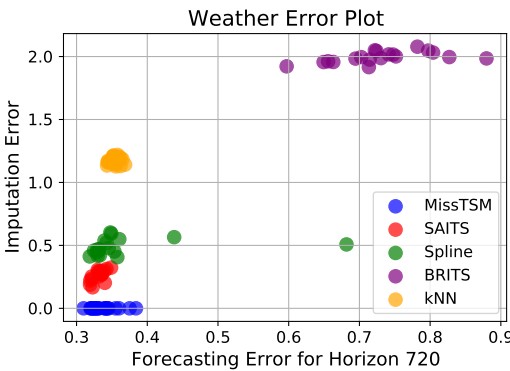

Figure 11: Imputation error vs Forecasting error across 5 trials for 4 missing fractions, 0.6, 0.7, 0.8, 0.9

## D  ANALYSIS OF IMPACT OF FREQUENCY AND PHASE PARAMETERS

In the following, we provide additional details regarding an ablation we conducted to understand the impact of frequency and phase parameters. Given the varying frequency and phase for each feature, we modify the intervals of both to assess their impact on the results. Dataset=ETTh2, Fraction=90%

**Case 1**. With the phase interval held constant, we lower the frequency range and examine two intervals: one in the high frequency region ([0.6, 0.9]) and one in the low frequency region ([0.1, 0.3]). The performance comparison between these new strategies and the original configuration is shown in Table 6.

Table 6: Effect of sampling from different frequency intervals. The best results are in bold and second-best are italicized

| Time Horizon | Original Periodic Masking MSE | High Frequency MSE | Low Frequency MSE |
|---|---|---|---|
| 96 | **0.268 ± 0.0151** | *0.281 ± 0.028* | 0.285 ± 0.023 |
| 192 | **0.295 ± 0.0298** | *0.301 ± 0.037* | 0.316 ± 0.049 |
| 336 | 0.319 ± 0.0185 | *0.308 ± 0.014* | **0.307 ± 0.011** |
| 720 | 0.356 ± 0.0310 | **0.339 ± 0.043** | *0.351 ± 0.058* |

We observe that with a reduced frequency range, for both high and low frequency intervals, the performance improves as the prediction window increases.

**Case 2**. Following a similar approach as Case 1, we keep the frequency interval constant and lower the range of phase values. We examine the following intervals: the positive half-cycle $[0, \pi]$ and the negative half-cycle $[\pi, 2\pi]$. Table 7 presents the results of this ablation

We observe a similar pattern here as well, with the performance improving as the prediction window increases when we sample from either the positive or negative cycle.

As shown in the tables above, frequency and phase values clearly impact model performance. The new strategies reduce frequency or phase-related randomness among the variates of the dataset, resulting in more consistent values. This appears to enhance the model's ability in long-term forecasting.

Table 7: Effect of sampling from different phase intervals. The best results are in bold and second-best are italicized

| Time Horizon | Original Periodic Masking MSE | (+) Half Cycle MSE | (-) Half Cycle MSE |
|---|---|---|---|
| 96 | **0.268 ± 0.0151** | *0.287 ± 0.037* | 0.293 ± 0.04 |
| 192 | **0.295 ± 0.0298** | *0.309 ± 0.05* | 0.313 ± 0.057 |
| 336 | 0.319 ± 0.0185 | *0.316 ± 0.022* | **0.311 ± 0.013** |
| 720 | 0.356 ± 0.0310 | *0.343 ± 0.035* | **0.340 ± 0.040** |

