# OpenReview forum: "Masking the Gaps: An Imputation-Free Approach to Time Series Modeling with Missing Data"
_ICLR.cc/2025/Conference — Submitted to ICLR 2025_

### Official Review · Reviewer_7Jfe · 2024-10-18

**Soundness:** 3
**Presentation:** 4
**Contribution:** 2
**Rating:** 8
**Confidence:** 3

**Summary:**

This work proposes an imputation-free approach for handling missing values in time series termed Missing Feature-aware Time
Series Modeling (MissTSM). MissTSM is composed of two primary modules. The first, Time-Feature Independent (TFI) Embedding, encodes each combination of time-step and feature into a high-dimensional space. The second module, the Missing Feature-Aware Attention (MFAA) Layer, is a novel mechanism designed to capture latent representations at each time-step by leveraging partially observed features. Empirical evaluations conducted under Masking Completely at Random (MCAR) and periodic masking validate the effectiveness of MissTSM in modeling time series data with missing features.

**Strengths:**

1. The intuition behind the idea is clear, effectively circumventing the imputation step typically used in traditional methods for handling missing data.
2. The presentation is strong, with clear figures that make the content easy to read and understand.
3. The experiments are comprehensive, covering both time series forecasting and classification, and include an insightful ablation study.

**Weaknesses:**

1. While the motivation is clear, the experimental results do not show significant improvement over the other baselines.
2. Time series missingness is a well-studied and widely researched field, making it difficult to assess the significance of the contribution here.
3. This work primarily represents an empirical study rather than a theoretical advancement.

**Questions:**

1. The author mentions two drawbacks of imputation: the propagation of imputation errors and the trade-off between imputation efficacy and complexity. Could the author clarify how MissTSM circumvents these drawbacks? I'm confused because: (1) it seems that errors in TFI would aggregate and influence MFAA, and (2) there's a clear trade-off between complexity and efficiency—for example, adding a dynamics module may improve performance but also increase complexity. Could the author clarify to what extent MissTSM addresses these two challenges?
2. Please clarify the intuition behind the 2D Positional Encodings (Line201-210) and explain why and how equations (2) and (3) are used in this context? Additionally, could the author elaborate on why these encodings provide an effective representation?
3. Can the authors explain the rationale for considering 60%-70% missing values in the experiments? Is it because MissTSM shows more significant improvements with a higher percentage of missing values? Typically, a missing range of 10-40% is more common, so additional clarification would be helpful.
4. My main concern lies in the significance of the experimental results. It appears that MissTSM's performance heavily depends on the dataset. For example, MissTSM only demonstrates comparable results on the ETTm2 and Weather datasets (Table 1, Figure 5), and the ablation study doesn’t clearly show the effectiveness of TFI and MFAA. Could the authors explain this? If additional results could be provided, specifically concerning MissTSM without TFI-Embedding but with MFAA, and address the previous questions, I would be happy to increase my score.

---

> ### Author Response · Authors · 2024-11-24
>
> **Comment 1: While the motivation is clear, the experimental results do not show significant improvement over the other baselines.**
> > As stated in our global response, MissTSM demonstrates consistently competitive performance across various datasets and excels in challenging scenarios, including high masking fractions and extended horizons. It surpasses baselines on real-world datasets like PhysioNet, underscoring its practical utility. Furthermore, its imputation-free design eliminates computational overhead and avoids the propagation of imputation errors into forecasting errors.
>
> **Comment 2: Time series missingness is a well-studied and widely researched field, making it difficult to assess the significance of the contribution here.**
>
> > There are two primary contributions of our work relative to other works in time-series modeling with missing values.
>
> >1. *Conceptually,*  MissTSM is the first adaptation of the idea of Masked Autoencoders (MAE) for modeling multivariate time-series with missing values. While there exists other adaptations of MAE for time-series data such as [1, 2], previous adaptations expect both training and testing datasets to be clean without missing values. This makes them limited in their applicability to real-world applications with varying data irregularities. In contrast, our method provides the flexibility to handle missing values directly within the framework of MAE, bringing the power of representation learning and masked pre-training to the domain of time-series modeling.
> 2. *Empirically,* we show that MissTSM shows *competitive performance* *consistently* across an extensive set of experiments involving forecasting and classification datasets. This when coupled with the fact that our method is imputation-free, makes it an ideal choice for handling missing values in time-series forecasting problems, because (a) it does not suffer from the computational costs of training complex imputation models such as SAITS, and (b) it does not involve propagation of imputation errors to forecasting errors as is the case for other baselines, as described in Appendix C.
>
> > [1] Dong, Jiaxiang, Haixu Wu, Haoran Zhang, Li Zhang, Jianmin Wang, and Mingsheng Long. "SimMTM: A simple pre-training framework for masked time-series modeling." Advances in Neural Information Processing Systems 36 (NeurIPS) 2023.
> [2] Li, Zhe, Zhongwen Rao, Lujia Pan, Pengyun Wang, and Zenglin Xu. "Ti-MAE: Self-supervised masked time series autoencoders." arXiv preprint arXiv:2301.08871 (2023).
>
> **Comment 3: This work primarily represents an empirical study rather than a theoretical advancement.**
>
> > We agree that our work primarily represents an empirical study rather than making theoretical contributions. However, empirical evaluations play a crucial role in advancing the field of time-series modeling, as reflected in several recent influential works focused on empirical contributions such as SimMTM [1] and iTransformer [2], published at NeurIPS 2023 and ICLR 2024, respectively. Our work contributes by systematically exploring the challenges of missing values in multivariate time-series data and proposing a novel approach to address these challenges that is the first adaptation of MAEs for handling missing values in time-series problems.
>
> > [1] Dong, Jiaxiang, Haixu Wu, Haoran Zhang, Li Zhang, Jianmin Wang, and Mingsheng Long. "SimMTM: A simple pre-training framework for masked time-series modeling." Advances in Neural Information Processing Systems 36 (NeurIPS) 2023
> [2] Liu, Yong, Tengge Hu, Haoran Zhang, Haixu Wu, Shiyu Wang, Lintao Ma, and Mingsheng Long. "iTransformer: Inverted transformers are effective for time series forecasting". International Conference on Learning Representations (ICLR) 2024

---

> > ### Author Response · Authors · 2024-11-24
> >
> > **Comment 4: The author mentions two drawbacks of imputation: the propagation of imputation errors and the trade-off between imputation efficacy and complexity. Could the author clarify how MissTSM circumvents these drawbacks? I'm confused because: (1) it seems that errors in TFI would aggregate and influence MFAA, and (2) there's a clear trade-off between complexity and efficiency—for example, adding a dynamics module may improve performance but also increase complexity. Could the author clarify to what extent MissTSM addresses these two challenges?**
> >
> > > We would like to clarify that MissTSM also involves a trade-off between model complexity and accuracy, which is universal to any machine learning system built on empirical risk minimization. MissTSM specifically avoids the trade-off between imputation complexity and efficacy faced by two-stage approaches, since MissTSM is imputation-free. MissTSM also does not involve propagation of imputation errors into the forecasting pipeline.
> >
> > > Regarding errors in TFI influencing MFAA, please note that we can only measure errors of our overall framework of MissTSM on downstream tasks as individual components such as TFI and MFAA are not being used for making predictions independently. As a result, we do not have a way to attribute errors on downstream tasks to either of the two components. of TFI and MFAA work with each together to handle missing values but none of them individually perform imputation or time-series modeling, encountering no trade-off between their objectives.
> >
> > **Comment 5: Please clarify the intuition behind the 2D Positional Encodings (Line 201-210) and explain why and how equations (2) and (3) are used in this context? Additionally, could the author elaborate on why these encodings provide an effective representation?**
> > > In TFI, each time-feature point is treated as a unique token with its own embedding. As a result the positional information for every token has to take into account, both, time and feature dimensions. To achieve this, we adopt 2D Positional Encodings, inspired by [1], which uniquely represent the position of each time-feature combination relative to other combinations. Equations (2) and (3) construct these encodings through sine and cosine functions that vary across the time and feature spaces, ensuring unique values for each combination. These encodings provide an effective representation because they allow the model to differentiate between TFI embeddings observed at different time and feature settings.
> >
> > > [1] Wang, Zelun, and Jyh-Charn Liu. "Translating math formula images to LaTeX sequences using deep neural networks with sequence-level training." International Journal on Document Analysis and Recognition (IJDAR) 24, no. 1 (2021): 63-75.
> >
> > **Comment 6: Can the authors explain the rationale for considering 60%-70% missing values in the experiments? Is it because MissTSM shows more significant improvements with a higher percentage of missing values? Typically, a missing range of 10-40% is more common, so additional clarification would be helpful.**
> >
> > > While many prior works have worked in moderate ranges of missing value fractions such as 10-40%, many real-world applications involve much higher levels of missing values. For example, PhysioNet, a real-world dataset considered in our experiments has 80% missing values. There are other time-series datasets in domains such as biomedicine and limnology that have even higher missing value fractions, owing to sensor failures, environmental obstructions, etc. We thus considered moderate to extreme ranges of missing value fractions in our experiments to closely resemble real-world scenarios, where we find MissTSM to be very effective under challenging conditions.
> >
> > **Comment 7: My main concern lies in the significance of the experimental results. It appears that MissTSM's performance heavily depends on the dataset. For example, MissTSM only demonstrates comparable results on the ETTm2 and Weather datasets (Table 1, Figure 5)**
> >
> > > As stated in the global comment, while MissTSM does not show best-performance across all scenarios, it consistently provides competitive results across varying datasets, horizon windows, masking schemes and fractions, with notable performance gains on larger missing fractions and longer horizon windows.

---

> > > ### Author Response · Authors · 2024-11-24
> > >
> > > **Comment 8: The ablation study doesn’t clearly show the effectiveness of TFI and MFAA. Could the authors explain this? Additional results could be provided for MissTSM without TFI-Embedding but with MFAA.**
> > >
> > > > Note that in our ablation study, removing MFAA layer automatically removes the TFI layer, because the TFI is not meaningful on its own without MFAA. Hence, our ablation study did not examine the individual effectiveness of these two components. Instead, it evaluated the combined effect of removing MFAA and TFI together. To clarify this, we have updated the captions of Figures 9 and 10 in the revised manuscript.
> > >
> > > > To understand the effect of removing TFI from MFAA, we conducted a simple ablation study where we fixed the weights of the TFI embedding layers to a constant value of 1 with a bias of 0, to simulate a scenario of no TFI embedding. This corresponds to using every time-feature combination as a scalar value rather than a multidimensional embedding. For this study, we considered the ETTh2 dataset across 60-90% MCAR missing fractions. The results are summarized in the Table below.
> > > | Fraction | Horizon | W/o TFI | With TFI |
> > > |----------|---------|---------|----------|
> > > | 60%      | 96      | 0.535   | **0.24**     |
> > > |          | 192     | 0.475   | **0.26**     |
> > > |          | 336     | 0.496   | **0.28**     |
> > > |          | 720     | 0.834   | **0.33**     |
> > > | 70%      | 96      | 0.419   | **0.25**     |
> > > |          | 192     | 0.458   | **0.27**     |
> > > |          | 336     | 0.583   | **0.30**     |
> > > |          | 720     | 1.730   | **0.35**     |
> > > | 80%      | 96      | 0.472   | **0.26**     |
> > > |          | 192     | 0.678   | **0.28**     |
> > > |          | 336     | 1.025   | **0.29**     |
> > > |          | 720     | 0.693   | **0.37**     |
> > > | 90%      | 96      | 0.500   | **0.32**     |
> > > |          | 192     | 0.677   | **0.35**     |
> > > |          | 336     | 1.050   | **0.32**     |
> > > |          | 720     | 0.665   | **0.35**     |
> > >
> > > > We can observe that MissTSM suffers significantly on removing the TFI embeddings, suggesting the importance of using learnable multidimensional embeddings for every time-feature combination rather than scalar values.

---

> > > > ### Comment · Reviewer_7Jfe · 2024-11-25
> > > >
> > > > Thanks for the authors' response. Could you further elaborate on the practical takeaways for using MissTSM? Specifically, when should this method be applied, and when should we consider other SOTA imputation techniques?

---

> > > > > ### Author Response · Authors · 2024-11-25
> > > > >
> > > > > Thank you for this suggestion. Here is a summary of the **practical takeaways** of using MissTSM compared to SOTA baselines with imputation techniques:
> > > > >
> > > > > 1. MissTSM shows *consistently competitive performance* across most dataset variations with varying synthetic masking schemes and masking fractions. MissTSM is especially effective in more challenging scenarios with **larger masking fractions** and **longer horizon windows** resembling real-world settings (e.g., PhysioNet).
> > > > >
> > > > > 2.  MissTSM is especially useful in resource-constrained applications where **computational costs need to be kept low**, since MissTSM is imputation-free and does not suffer from the complexity-accuracy tradeoffs of SOTA imputation techniques. As a point of reference, MissTSM uses only 28K parameters on ETTh2 while SOTA baselines such as SimMTM, iTransformer, and PatchTST require 4.69M, 254K, and 81K parameters, respectively, in addition to the imputation modeling overhead.
> > > > >
> > > > > 3. MissTSM does not currently beat SOTA baselines in simpler scenarios with lower masking fractions and shorter horizon windows in terms of prediction performance (as seen in our results on Weather and EMG datasets). While SOTA methods may appear more effective in such scenarios, we would like to emphasize that MissTSM represents the first work in a new line of methods for time-series modeling that are *fast, imputation-free, and modeling assumption-free* based on the idea of masked auto-encoders. We anticipate MissTSM to inspire future directions in the same conceptual framework to address its current limitations, e.g., by extending MissTSM to jointly learn cross-channel correlations along with non-linear temporal dynamics.

---

> > > > > > ### Comment · Reviewer_7Jfe · 2024-11-25
> > > > > >
> > > > > > Thanks for the response. I keep a positive view of this work, but I strongly encourage the authors to carefully consider all reviewers' suggestions in the revised manuscript.

---

> > > > > > > ### Author Response · Authors · 2024-11-25
> > > > > > >
> > > > > > > Thank you for your valuable input. We appreciate your suggestions and will ensure that we carefully address all reviewers' comments in the revised manuscript.

---

### Official Review · Reviewer_G7Mn · 2024-11-01

**Soundness:** 2
**Presentation:** 1
**Contribution:** 2
**Rating:** 3
**Confidence:** 4

**Summary:**

To address the missing values in time series datasets, this work considers the imputation-free time series modeling. MCAR and periodic masking are considered in experiments by comparing with two stage baselines, consisting of imputation and modeling approaches.

**Strengths:**

S1. Various imputation baselines are considered in experiments.

S2. Time series modeling is important.

S3. Time series forecasting and classification are considered in experiments.

**Weaknesses:**

W1. The introduction is not well-written, which seems more like the related work. It is suggested to further polish the manuscript, to improve the readability.

W2. In the Introduction, the authors claim that all existing works for time-series modeling with missing values are two stage approaches, where missing values are first imputed, followed by feeding the imputed time-series to a modeling approach. Unfortunately, there already exists existing study [m1] considering the robust time series modeling over low quality data, where missing values and anomalies are associated with low importance in modeling, instead of treated as a two stage framework. Therefore, the motivation is unpersuasive, claiming that all the existing works are two stage approaches.

W3. In practice, in addition to the missing values, there also exist various data quality problems, e.g., anomaly, errors, staleness, etc. Only focusing modeling the time series data with missing values is insufficient to serve real applications. Please discuss how the approach might be extended or adapted to handle other types of data quality problems, or explain why you chose to focus specifically on missing values.

W4. MCAR and periodic masking are widely used operations in time series imputation, which are surprising to be proposed by this manuscript, at the end of the Introduction.

W5. Various imputation methods are considered in the experiments, but the most related works, i.e., robust modeling methods such as [m1], are totally ignored.

W6. The SOTA imputation baselines are ignored in the comparison, such as [m2]. Please include the comparison with it.

W7. In addition to the MCAR and periodic masking, MAR and MNAR are also widely considered in the experiments.

[m1] Cheng, Hao, et al. "RobustTSF: Towards Theory and Design of Robust Time Series Forecasting with Anomalies." The Twelfth International Conference on Learning Representations.

[m2] Tashiro, Yusuke, et al. "Csdi: Conditional score-based diffusion models for probabilistic time series imputation." Advances in Neural Information Processing Systems 34 (2021): 24804-24816.

**Questions:**

W1-W5

---

> ### Author Response · Authors · 2024-11-24
>
> **Comment 1: The introduction is not well-written, which seems more like the related work. It is suggested to further polish the manuscript, to improve the readability.**
>
> > We apologize if our Introduction section was hard to read. We will reduce the discussions of related works and move them to the next section to improve the readability of this section.
>
> **Comment 2: There already exists single-stage frameworks such as RobustTSF that perform time series modeling with low quality data, including missing values and anomalies. Therefore, the motivation is unpersuasive, claiming that all the existing works are two-stage approaches. RobustTSF should be included in the experiments.**
>
> > Thank you for pointing us to the impactful work of RobustTSF, which we are happy to include in our related works discussion. There are four key differences between RobustTSF and MissTSM.
> > 1. RobustTSF is a **filtering** approach to ignore samples with anomalies or missing values (treated as anomalies after being imputed with zero values) during training to ensure robustness. In contrast, MissTSM focuses on **representation learning** to learn latent representations of observed variates at every time-point, without dropping time-points with missing values. MissTSM thus leverages all of the available information of observed features across time while only ignoring missing time-feature pairs.
> 2. RobustTSF makes several simplifying assumptions to identify anomalies, including trend filtering based on smoothness assumptions and anomaly score computation based on Dirac weighting functions, which may not be held in real-world applications. These assumptions additionally require user-input parameters such as anomaly score threshold. In contrast, MissTSM makes no such modeling assumption and completely relies on the supervision contained in observed data.
> 3. While RobustTSF is designed to handle missing values during training, it assumes that there are no missing values during testing. As a result, RobustTSF is unable to handle missing values in the inputs during testing beyond simply imputing it with zero values. In contrast, MissTSM can handle missing values both during training and testing.
> 4. RobustTSF is formulated for univariate time-series problems. Even though it is possible to apply RobustTSF in multivariate settings, it treats each variate independently of other variates and does not take into account interactions among variates. In contrast, MissTSM is designed for multivariate time-series modeling.
>
> > To compare RobustTSF with MissTSM, we consider a multivariate forecasting experiment over the ETTh2 dataset across 60% to 90% MCAR missing data fractions. We use the Autoformer model as the time-series model integrated with RobustTSF
> | Fraction | Horizon | RobustTSF + Autoformer   | MissTSM          |
> |----------|---------|--------------------------|------------------|
> | 60%      | 96      | 1.5 ± 0.054             | **0.24 ± 0.007**     |
> |          | 192     | 1.45 ± 0.065            | **0.26 ± 0.002**     |
> |          | 336     | 1.34 ± 0.082            | **0.28 ± 0.009**     |
> |          | 720     | 1.33 ± 0.024            | **0.33 ± 0.012**     |
> | 70%      | 96      | 1.81 ± 0.117            | **0.25 ± 0.009**     |
> |          | 192     | 1.76 ± 0.056            | **0.27 ± 0.013**     |
> |          | 336     | 1.76 ± 0.049            | **0.30 ± 0.008**     |
> |          | 720     | 1.69 ± 0.032            | **0.35 ± 0.01**      |
> | 80%      | 96      | 2.2 ± 0.084             | **0.26 ± 0.025**     |
> |          | 192     | 2.23 ± 0.091            | **0.28 ± 0.021**     |
> |          | 336     | 2.2 ± 0.085             | **0.29 ± 0.029**     |
> |          | 720     | 2.11 ± 0.041            | **0.37 ± 0.056**     |
> | 90%      | 96      | 2.72 ± 0.077            | **0.32 ± 0.029**     |
> |          | 192     | 2.74 ± 0.036            | **0.35 ± 0.072**     |
> |          | 336     | 2.77 ± 0.024            | **0.32 ± 0.014**     |
> |          | 720     | 2.67 ± 0.034            | **0.35 ± 0.045**     |
>
> > We observe that MissTSM clearly outperforms RobustTSF + Autoformer across all the missing fractions and prediction horizon, thus highlighting the impact of our contribution in relation to this prior work.

---

> > ### Author Response · Authors · 2024-11-24
> >
> > **Comment 3: Only focusing on missing values while modeling time-series is insufficient to serve real applications. Please discuss how the approach might be extended or adapted to handle other types of data quality problems, or explain why you chose to focus specifically on missing values.**
> >
> > > We agree that missing values is just one type of data quality problem and addressing a broader range of data quality problems (e.g., missing values with anomalies) is a valuable direction for future research. However, handling a single data quality problem is in itself very challenging for multivariate time-series modeling, let alone their combination. While methods such as RobustTSF try to address multiple data quality problems, they make several simplifying assumptions and involve pre-processing steps such as imputing missing values with zero that may not always be valid (e.g., when observed data naturally has values close to 0). We thus concentrate our efforts to handling missing values in multivariate time-series, which are quite prevalent in real-world applications.
> >
> > > While our focus is on missing values, MissTSM can be easily extended to handle anomalies by coupling it with off-the-shelf algorithms designed to detect anomalies in time-series data such as [1,2,3]. In particular, if we can detect anomalous time-feature pairs using existing algorithms, we can treat them as missing values in MissTSM, such that the training and testing of time-series models will only be impacted by non-anomalous observations. This extension can be explored in future works.
> >
> > >[1] Munir, Mohsin, Shoaib Ahmed Siddiqui, Andreas Dengel, and Sheraz Ahmed. "DeepAnT: A deep learning approach for unsupervised anomaly detection in time series." IEEE Access 7 (2018): 1991-2005.
> > [2] Su, Ya, Youjian Zhao, Chenhao Niu, Rong Liu, Wei Sun, and Dan Pei. "Robust anomaly detection for multivariate time series through stochastic recurrent neural network." In Proceedings of the 25th ACM SIGKDD International Conference on Knowledge Discovery & Data Mining, pp. 2828-2837. 2019.
> > [3] Xu, Jiehui. "Anomaly transformer: Time series anomaly detection with association discrepancy." arXiv preprint arXiv:2110.02642 (2021).
> >
> > **Comment 4: MCAR and periodic masking are widely used operations in time series imputation, which are surprising to be proposed by this manuscript, at the end of the Introduction.**
> >
> > > We apologize for the lack of clarity in our wording. We did not intend to claim these masking operations as novel contributions of our work. We will revise the sentence to highlight that we are using these operations that are well-known in the community to create novel versions of the benchmark datasets through synthetic masking, to simulate varying scenarios of missing values.

---

> ### Comment · Reviewer_G7Mn · 2024-11-26
> **Thanks for responses**
>
> Since ``MissTSM can be easily extended to handle anomalies by coupling it with off-the-shelf algorithms'', can you show the specific solution with the experimental results, compared with RobustTSF?
> In addition, why only high missing rates 60%-90% are considered? How about the experimental results for lower missing rates?

---

> > ### Author Response · Authors · 2024-11-30
> >
> > Thank you for the additional comments. Here are our responses to these comments.
> >
> > **Comment 1: Since “MissTSM can be easily extended to handle anomalies by coupling it with off-the-shelf algorithms”, can you show the specific solution with the experimental results, compared with RobustTSF?**
> >
> > > Certainly, we have now conducted new experiments to show the applicability of MissTSM for handling anomalies while modeling time-series on the Pooled Server Metrics (PSM) [1] dataset using the following setup. We first use an off-the-shelf anomaly detection method, Anomaly Transformer [2], to identify anomalies in the input dataset, which are then treated as missing values in the training of MissTSM. We call this combined framework: Anomaly Transformer + MissTSM. We use 134K samples for training the Anomaly Transformer and use the remaining 84K samples for training and evaluating two forecasting models, namely, Anomaly Transformer + MissTSM, and RobustTSF (with AutoFormer backbone). We specifically split the 84K samples into train, test, and validation sets using a 7:2:1 ratio. Once trained, we evaluate both MissTSM and RobustTSF on the test set ignoring anomalous points in the forecasting window (using ground-truth labels of anomalies). Here are the results of this experiment:
> > | Horizon Window | RobustTSF+Autoformer (MSE) | Anomaly Transformer + MissTSM (MSE) |
> > |-----------------|----------------------------|---------------|
> > | 96 | **0.349** | 0.360 |
> > | 192 | 0.524 | **0.465** |
> > | 336 | 0.731 | **0.603** |
> > | 720 | 0.958 | **0.815** |
> >
> > > We can see that Anomaly Transformer + MissTSM performs comparable or even better than RobustTSF across most horizon windows. The specific implementation of our new framework is present in our anonymous Github link here: https://anonymous.4open.science/r/MissTSM-2-ICLR-64CE/.  We would like to emphasize that while handling anomalies is not the primary focus of our work, this experiment demonstrates that our framework can be easily adapted to handle other data quality issues such as anomalies, further bolstering the significance of our contribution.
> >
> > >[1] Abdulaal, Ahmed, Zhuanghua Liu, and Tomer Lancewicki. "Practical approach to asynchronous multivariate time series anomaly detection and localization." In Proceedings of the 27th ACM SIGKDD conference on knowledge discovery & data mining, pp. 2485-2494. 2021.
> > [2] Xu, Jiehui. "Anomaly transformer: Time series anomaly detection with association discrepancy." arXiv preprint arXiv:2110.02642 (2021).
> >
> > **Comment 2: In addition, why only high missing rates 60%-90% are considered? How about the experimental results for lower missing rates?**
> >
> > > Thank you for this comment. As stated in our response to *Reviewer 7Jfe*, here is the justification for using high missing value fractions in our experiments.
> >
> > > *“While many prior works have worked in moderate ranges of missing value fractions such as 10-40%, many real-world applications involve much higher levels of missing values. For example, PhysioNet, a real-world dataset considered in our experiments has 80% missing values. There are other time-series datasets in domains such as biomedicine and limnology that have even higher missing value fractions, owing to sensor failures, environmental obstructions, etc. We thus considered moderate to extreme ranges of missing value fractions in our experiments to closely resemble real-world scenarios, where we find MissTSM to be especially very effective under challenging conditions.”*
> >
> > > Additionally, we compare the performance of RobustTSF and MissTSM on smaller missing value fractions (10% to 40%) in the table below.
> > | **Fraction** | **Horizon** | **RobustTSF+ Autoformer** | **MissTSM** |
> > |--------------|-------------|-------------------------------|------------------------------|
> > | **10%** | 96 | 0.41 ± 0.014 | **0.25 ± 0.005** |
> > | | 192 | 0.42 ± 0.015 | **0.29 ± 0.05** |
> > | | 336 | 0.44 ± 0.029 | **0.32 ± 0.016** |
> > | | 720 | 0.47 ± 0.02 | **0.33 ± 0.013** |
> > | **20%** | 96 | 0.47 ± 0.009 | **0.25 ± 0.004** |
> > | | 192 | 0.47 ± 0.008 | **0.26 ± 0.008** |
> > | | 336 | 0.47 ± 0.013 | **0.31 ± 0.017** |
> > | | 720 | 0.51 ± 0.012 | **0.32 ± 0.026** |
> > | **30%** | 96 | 0.61 ± 0.017 | **0.27 ± 0.01** |
> > | | 192 | 0.60 ± 0.011 | **0.28 ± 0.008** |
> > | | 336 | 0.59 ± 0.016 | **0.31 ± 0.002** |
> > | | 720 | 0.60 ± 0.012 | **0.34 ± 0.002** |
> > | **40%** | 96 | 0.80 ± 0.038 | **0.26 ± 0.011** |
> > | | 192 | 0.78 ± 0.036 | **0.27 ± 0.018** |
> > | | 336 | 0.76 ± 0.045 | **0.34 ± 0.05** |
> > | | 720 | 0.77 ± 0.015 | **0.35 ± 0.02** |
> >
> > > We can see that MissTSM consistently outperforms RobustTSF even at lower missing fractions.

---

### Official Review · Reviewer_UWVS · 2024-11-03

**Soundness:** 2
**Presentation:** 3
**Contribution:** 1
**Rating:** 3
**Confidence:** 5

**Summary:**

This paper introduces a DL-based time-series method called MissTSM, which is designed to handle missing values in time series data without relying on imputation techniques. The key components of MissTSM include a Time-Feature Independent (TFI) embedding that treats each time-step and feature combination as a distinct token and a Missing Feature-Aware Attention (MFAA) layer that learns latent representations at each time-step based on the observed features (using missingness indicator when assigning the attention values). The authors evaluate MissTSM on multiple benchmark datasets using synthetic (MCAR and periodic) and real-world missing value scenarios. The results show that MissTSM achieves comparable performance to a wide variety of time-series classification and forecasting methods (equipped with state-of-the-art imputation methods).

**Strengths:**

-	The paper is in general well-written and easy to follow.

**Weaknesses:**

-	The authors have presented a limited perspective on time-series methods that avoid imputation. A broader range of approaches exists to tackle missing values in time series. One such approach involves augmenting the input data with indicators for each feature-time variable, which can be particularly informative when missingness is not random. Another line of research focuses on handling multivariate time series with irregular time intervals. This includes methods like GRU-D [A], which directly incorporates missingness indicators and time intervals as inputs, and ODE/SDE-based approaches [B]-[D] that inherently model such irregular time series.
-	The authors should consider a broader range of baseline models. This could include evaluating existing time-series methods that incorporate missingness indicators as input or exploring some of the mentioned techniques designed for handling irregular time intervals.
-	Treating each feature at each time point as a separate embedding and treating missingness indicator to bias the attention score are not technically novel. Moreover, the performance gain seems marginal and sometimes being lot worse than the benchmarks.
-	While the utility of the imputation methods is often measured by the performance in downstream tasks, imputation is not designed to achieve higher discriminative/prediction power. More simpler imputation methods – such as mean/mode imputation or last observation carried forward – should be included as well in the experiments.

References
- [A] Che et al., “Recurrent Neural Networks for Multivariate Time Series with Missing Values,” Scientific Reports, 2018.
- [B] Chen et al., “Neural ordinary differential equations,” NeurIPS 2018.
- [C] Rubanova et al., “Latent ODEs for Irregularly-Sampled Time Series,” NeurIPS 2019.
- [D] Park et al., “Neural Stochastic Differential Games for Time-series Analysis,” ICML 2023.

**Questions:**

-	It seems that attention scores for the masked feature-time points can be naturally removed when computing those self-attention scores. What is the benefit of using the mask indicators as the bias terms rather than explicitly removing those from computation?
-	Why the quarries do not depend on the time point whereas the keys and values do?

---

> ### Author Response · Authors · 2024-11-24
>
> **Comment 1: The authors should consider a broader range of baseline models, including methods that incorporate missingness indicators such as GRU-D, or methods that handle irregular time intervals using ODE/SDE-based approaches such as Neural ODE, Latent ODE, and Neural Stochastic Differential Games for Time-series Analysis**
> > We thank the reviewer for suggesting these prior works [1,2,3,4] that we are happy to include in our discussion of related works. While prior works such as GRU-D [4] embed time intervals between observations as auxiliary features to handle irregular time-sequences, our approach aims to bypass this step and learn representations based on the observed data along both time and feature dimensions. Also, while ODE-based prior works [1,2] offer an elegant framework for modeling irregularly sampled data in continuous time, they rely on solving differential equations that can be computationally demanding and difficult to scale, as experienced in our experiment runs. To include these prior works in our comparisons of results, we have performed new experiments with GRU-D and Latent ODE for both classification and forecasting tasks, as summarized below.
>
> > *Comparison of MissTSM with GRU-D on Classification tasks:* We use the available GRU-D re-implementation (as the official code is not available currently) and adapt it for the classification task. The model was run for 100 epochs. We consider an MCAR setup with a missing fraction of 80%. Datasets considered are Epilepsy, Gesture, and EMG.
> | Dataset | GRU-D (F1) | MissTSM (F1) |
> |-----------|------------|--------------|
> | Epilepsy | 6.52% | **64.9%** |
> | Gesture | 3.16% | **55.70%** |
> | EMG | 2.78% | **59.45%** |
>
> > *Comparison of MissTSM with LatentODE on ETTh2:* We consider Latent ODE with ODE-RNN as the encoder. Since ODE-based methods are very computationally expensive, we consider a simple setup of 336 context length and 96 prediction length. For simplicity, we only consider MCAR masking with varying masking fractions.
> | Fraction | Latent ODE (MSE) | MissTSM (MSE) |
> |----------|------------------------|--------------------|
> | 60% | 4.25 | **0.243** |
> | 70% | 3.181 | **0.250** |
> | 80% | 2.543 | **0.264** |
> | 90% | 2.624 | **0.316** |
>
> > We can see from both these Tables that MissTSM performs significantly better than GRU-D and Latent ODE, highlighting the impact of our contribution in relation to prior works in time-series modeling with missing values.
>
> >[1] Chen, Ricky TQ, Yulia Rubanova, Jesse Bettencourt, and David K. Duvenaud. "Neural ordinary differential equations." Advances in neural information processing systems 31 (2018)
> [2] Rubanova, Yulia, Ricky TQ Chen, and David K. Duvenaud. "Latent ordinary differential equations for irregularly-sampled time series." Advances in neural information processing systems 32 (2019)
> [3] Park, Sungwoo, Byoungwoo Park, Moontae Lee, and Changhee Lee. "Neural stochastic differential games for time-series analysis." International Conference on Machine Learning (2023)
> [4] Che, Zhengping, Sanjay Purushotham, Kyunghyun Cho, David Sontag, and Yan Liu. "Recurrent neural networks for multivariate time series with missing values." Scientific reports 8, no. 1 (2018): 6085.

---

> > ### Author Response · Authors · 2024-11-24
> >
> > **Comment 2: Treating each feature at each time point as a separate embedding and treating missingness indicator to bias the attention score are not technically novel.**
> >
> > > While our work builds upon familiar architectural components of transformers such as attentions, the **core novelty of our work** lies in how we rethink attention mechanisms to handle missing values in time-series data, inspired by similar ideas in vision and language domains such as masked auto-encoders (MAE) [1]. Specifically, the novelty of our work lies in how we use masked attentions to learn feature representations of multiple variates with missing values via a novel framework of Missing Feature Aware Attention (MFAA). MFAA requires every time-feature pair to be treated as a separate token with its own embedding, which we achieve through a novel Time-Feature Independent (TFI) embedding module. Together, MFAA and TFI represent the novel contributions of our work that when embedded in an encoder-decoder framework results in the proposed architecture of MissTSM, as shown in Figure 3 of the main paper.
> >
> > > We would also like to emphasize that other influential works such as iTransformer [2] published at ICLR 2024 also build upon the same architectural components as Transformers but apply these components in innovative ways for time-series modeling. Our work represents a novel use of attention mechanisms to enable representation learning of multivariate time-series in the presence of missing values, extending the idea of masked auto-encoders to the domain of time-series modeling.
> >
> > > [1] He, Kaiming, Xinlei Chen, Saining Xie, Yanghao Li, Piotr Dollár, and Ross Girshick. "Masked autoencoders are scalable vision learners." In Proceedings of the IEEE/CVF conference on computer vision and pattern recognition, pp. 16000-16009. 2022.
> > [2] Liu, Yong, Tengge Hu, Haoran Zhang, Haixu Wu, Shiyu Wang, Lintao Ma, and Mingsheng Long. "iTransformer: Inverted transformers are effective for time series forecasting. International Conference on Learning Representations (ICLR) 2024
> >
> > **Comment 3: The performance gains of MissTSM seems marginal and sometimes being lot worse than the baselines.**
> >
> > > As discussed in our global response, MissTSM consistently shows competitive performance across diverse datasets and shows significant performance gains in challenging scenarios such as high masking fractions and large horizon windows. MissTSM also significantly outperforms baselines on a real-world dataset, Physionet, showcasing its practical applicability. Moreover, the imputation-free nature of our model prevents the computational overhead and propagation of imputation error to forecasting error.
> >
> > **Comment 4: More simpler imputation methods – such as mean/mode imputation or last observation carried forward (LOCF) – should be included as well in the experiments.**
> >
> > > We thank the reviewer for this suggestion. We have generated new results of the LOCF imputation method when coupled with the iTransformer model on the ETTm2 dataset, summarized in the Table below (reported values are MSE).
> > | Horizon | Fraction | iTransformer + LOCF | MissTSM |
> > |---------|----------|----------------------------|--------------------------|
> > | 96 | 60% | **0.18 +/- 0.002** | 0.22 +/- 0.005 |
> > | |      70% | **0.18 +/- 0.002** | 0.23 +/- 0.006 |
> > | |      80% | **0.19 +/- 0.0022** | 0.23 +/- 0.012 |
> > | |      90% | **0.21 +/- 0.0023** | 0.24 +/- 0.034 |
> > |192 | 60% | **0.25 +/- 0.003**  | **0.25 +/- 0.009** |
> > | |        70% | **0.26 +/- 0.002**  | 0.26 +/- 0.018 |
> > | |        80% | **0.25 +/- 0.0025** | 0.27 +/- 0.013 |
> > | |        90% | **0.27 +/- 0.003** | **0.27 +/- 0.010** |
> > | 336 | 60% | 0.30 +/- 0.006 | **0.29 +/- 0.019** |
> > | |         70% | **0.30 +/- 0.002** | **0.30 +/- 0.012** |
> > | |         80% | 0.31 +/- 0.0044 | **0.28 +/- 0.010** |
> > | |         90% | **0.32 +/- 0.005** | **0.32 +/- 0.043** |
> > | 720 | 60% | 0.38 +/- 0.002 | **0.31 +/- 0.014** |
> > | |         70% | 0.38 +/- 0.0019 | **0.31 +/- 0.013** |
> > | |         80% | 0.38 +/- 0.0035 | **0.32 +/- 0.020** |
> > | |         90% | 0.39 +/- 0.006 | **0.34 +/- 0.024**|
> >
> > > We can see that MissTSM performs better than the baseline using LOCF especially for larger horizon windows, which is a trend consistent with the other results in the paper (see Figure 5). In Figure 6 of the main paper, we have also included results of baseline models iTransformer and PatchTST when coupled with the kNN imputation method, which is related to the mean imputation method suggested by the reviewer but applied in local neighborhoods.

---

> > > ### Author Response · Authors · 2024-11-24
> > >
> > > **Comment 5: It seems that attention scores for the masked feature-time points can be naturally removed when computing those self-attention scores. What is the benefit of using the mask indicators as the bias terms rather than explicitly removing those from computation?**
> > >
> > > >We borrow the idea of using mask indicators as bias terms from BERT [1], which is now a standard technique commonly used in attention-based models.
> > >
> > > >[1] Devlin, Jacob, Ming-Wei Chang, Kenton Lee, and Kristina Toutanova. "BERT: Pre-training of Deep Bidirectional Transformers for Language Understanding." arXiv preprint arXiv:1810.04805 (2019)
> > >
> > > **Comment 6: Why do the queries not depend on the time point whereas the keys and values do?**
> > >
> > > > This separation of roles is inspired by similar architectures in multi-modal grounding [1, 2, 3], where a query from one modality (e.g., text) is used to attend to relevant regions in another modality (e.g., an image). For example, in DETR [1], learnable object queries are kept independent of the image content that is sent as keys and values. Another example is ClimaX [4] where learnable queries are used for variable aggregation that do not change with time. In our setting, the learnable queries capture the interactions among variates independent of time, enabling the model to attend to the most informative aspects of observed variates at any time-step fed through keys and values. This allows the model to focus on learning time-independent embeddings of all features as queries that can be applied to keys and values observed at every time-step.
> > >
> > > > [1] Carion, Nicolas, Francisco Massa, Gabriel Synnaeve, Nicolas Usunier, Alexander Kirillov, and Sergey Zagoruyko. "End-to-end object detection with transformers." In European conference on computer vision (ECCV) 2020.
> > > [2] Peng, Zhiliang, Wenhui Wang, Li Dong, Yaru Hao, Shaohan Huang, Shuming Ma, and Furu Wei. "Kosmos-2: Grounding multimodal large language models to the world." arXiv preprint arXiv:2306.14824 (2023).
> > > [3] Shi, Fengyuan, Ruopeng Gao, Weilin Huang, and Limin Wang. "Dynamic mdetr: A dynamic multimodal transformer decoder for visual grounding." IEEE Transactions on Pattern Analysis and Machine Intelligence (2023).
> > > [4] Nguyen, Tung, Johannes Brandstetter, Ashish Kapoor, Jayesh K. Gupta, and Aditya Grover. "ClimaX: A foundation model for weather and climate." arXiv preprint arXiv:2301.10343 (2023).

---

> ### Comment · Reviewer_UWVS · 2024-11-25
> **Response to the authors' rebuttal**
>
> I appreciate the authors taking the time to address my concerns in their rebuttal. However, I still have reservations about the technical novelty of this work, particularly in light of the new experiments comparing it to LoCF. While the finding that MissTSM performs better only over longer time horizons is intriguing, I believe a more thorough discussion of its advantages and disadvantages compared to simpler imputation methods is necessary.
>
> Additionally, I'm unfamiliar with the concept of biasing attention scores using mask indicators in BERT, as mentioned by the authors. To my knowledge, BERT utilizes [mask] tokens, and I'm not aware of any mechanism for directly biasing attention scores in this way. (I would be grateful if the authors could clarify this point and correct me if I'm mistaken.) Furthermore, I think the authors need to explain the reasoning behind this biasing approach more clearly. The impact of such a bias could vary significantly based on hyperparameter choices, and it seems like those choices would be heavily dependent on the specific temporal dynamics of the data. This aspect requires further elaboration.
>
> While I appreciate the authors' efforts, I don't believe this paper currently meets the high standards of ICLR. Therefore, I will maintain my original score.

---

> > ### Author Response · Authors · 2024-11-27
> >
> > Thank you for the additional comments. Here are our responses to these comments.
> >
> > **Comment 1: I still have reservations about the technical novelty of this work, particularly in light of the new experiments comparing it to LoCF.**
> >
> > > We agree that it is intriguing to see LOCF performing so well on the ETTm2 dataset despite its simplicity. To provide more points for comparison, we have revised the Table to include results with other imputation methods such as spline and SAITS as well, as shown below.
> > | Horizon | Fraction | iTransformer + LOCF | iTransformer + Spline | iTransformer + SAITS | MissTSM |
> > |---------|----------|----------------------------|--------------------------|----------------|---------------|
> > | 96 | 60% | **0.18 +/- 0.002** | **0.18 +/- 0.002** | 0.37 +/- 0.08 | 0.22 +/- 0.005 |
> > | |      70% | **0.18 +/- 0.002** | **0.18 +/- 0.006** | 0.46 +/- 0.11 | 0.23 +/- 0.006 |
> > | |      80% | 0.19 +/- 0.0022 | **0.18 +/- 0.008** | 0.64 +/- 0.14| 0.23 +/- 0.012 |
> > | |      90% | 0.21 +/- 0.0023 | **0.20 +/- 0.009** | 0.86 +/- 0.22 | 0.24 +/- 0.034 |
> > |192 | 60% | 0.25 +/- 0.003  | **0.24 +/- 0.005** | 0.43 +/- 0.08 | 0.25 +/- 0.009 |
> > | |        70% | 0.26 +/- 0.002  | **0.25 +/-  0.007** | 0.54 +/- 0.14 | 0.26 +/- 0.018 |
> > | |        80% | **0.25 +/- 0.0025** | **0.25 +/- 0.005** | 0.69 +/- 0.13 | 0.27 +/- 0.013 |
> > | |        90% | **0.27 +/- 0.003** | **0.27 +/- 0.01** | 0.93 +/- 0.19 | **0.27 +/- 0.010** |
> > | 336 | 60% | 0.30 +/- 0.006 | 0.29 +/-  0.002 | 0.48 +/- 0.08 | **0.29 +/- 0.019** |
> > | |         70% | **0.30 +/- 0.002** | **0.30 +/ 0.009** | 0.61 +/- 0.17 | **0.30 +/- 0.012** |
> > | |         80% | 0.31 +/- 0.0044 | 0.29 +/ 0.006 | 0.73 +/- 0.13 | **0.28 +/- 0.010** |
> > | |         90% | **0.32 +/- 0.005** | **0.32 +/- 0.008** | 1.04 +/ 0.18 | **0.32 +/- 0.043** |
> > | 720 | 60% | 0.38 +/- 0.002 | 0.38 +/-  0.008 | 0.54 +/- 0.07 | **0.31 +/- 0.014** |
> > | |         70% | 0.38 +/- 0.0019 | 0.38 +/- 0.008 | 0.64 +/- 0.14 | **0.31 +/- 0.013** |
> > | |         80% | 0.38 +/- 0.0035 | 0.39 +/- 0.009 | 0.80 +/- 0.16 | **0.32 +/- 0.020** |
> > | |         90% | 0.39 +/- 0.006 | 0.40 +/- 0.01 | 1.00 +/- 0.21| **0.34 +/- 0.024**|
> >
> > >We can see that simpler imputation methods such as LOCF and spline are consistently beating the more complex SAITS across all horizon windows of this dataset. This raises questions on the usefulness of complex imputation methods such as SAITS and the practical validity of trends observed on this simple dataset in resembling real-world performance.
> > To show the usefulness of SAITS compared to LOCF, the SAITS paper [1] already compared the performance of both methods using a standard baseline (RNN classifier) on the PhysioNet dataset, a challenging real-world dataset containing 80% missing values. We compare the F1-score of MissTSM with the other two methods on PhysioNet in the Table below.
> > | LOCF | SAITS | MissTSM |
> > |---------|----------|----------|
> > |39.5 % | 42.7 %| **57.84 %**|
> >
> > >We can see that on this more challenging dataset, LOCF no longer performs better than SAITS, and MissTSM is significantly better than both LOCF and SAITS. This demonstrates the practical usability of MissTSM in handling missing values in complex real-world scenarios beyond simplistic benchmark datasets with synthetic masking schemes.
> >
> > > [1] Du, Wenjie, David Côté, and Yan Liu. "Saits: Self-attention-based imputation for time series." Expert Systems with Applications 219 (2023): 119619.

---

> ### Author Response · Authors · 2024-11-27
>
> **Comment 2: I believe a more thorough discussion of its advantages and disadvantages compared to simpler imputation methods is necessary.**
>
> >Thank you for this suggestion. Here is a summary of the **practical takeaways** of using MissTSM compared to SOTA baselines with simpler imputation techniques:
> >1. MissTSM shows *consistently competitive performance* across most dataset variations with varying synthetic masking schemes and masking fractions. MissTSM is especially effective in more challenging scenarios with **larger masking fractions** and **longer horizon windows** resembling real-world settings (e.g., PhysioNet).
> 2.  MissTSM is especially useful in resource-constrained applications where **computational costs need to be kept low**, since MissTSM is imputation-free and does not suffer from the complexity-accuracy tradeoffs of SOTA imputation techniques. While simpler imputation methods may have low computational overhead, they suffer from the complexity-accuracy trade-off, as seen from the performance on PhysioNet.
> 3. MissTSM does not currently beat simple imputation-based baselines in simpler scenarios like shorter horizon windows in terms of prediction performance (as seen on the ETTm2 dataset above). While simple imputations may appear more effective in such scenarios, we would like to emphasize that the performance gap is relatively small compared to the gain in improvement achieved over more complex scenarios. Moreover, MissTSM represents the first work in a new direction of methods for time-series modeling that are *fast, imputation-free, and modeling assumption-free*. We anticipate MissTSM to inspire future directions in the same conceptual framework as masked auto-encoders to address its current limitations, e.g., by extending MissTSM to jointly learn cross-channel correlations along with non-linear temporal dynamics.
>
> **Comment 3: I'm unfamiliar with the concept of biasing attention scores using mask indicators in BERT, as mentioned by the authors. To my knowledge, BERT utilizes [mask] tokens, and I'm not aware of any mechanism for directly biasing attention scores in this way. (I would be grateful if the authors could clarify this point and correct me if I'm mistaken.) Furthermore, I think the authors need to explain the reasoning behind this biasing approach more clearly. The impact of such a bias could vary significantly based on hyperparameter choices, and it seems like those choices would be heavily dependent on the specific temporal dynamics of the data. This aspect requires further elaboration**
>
> > Apologies for the misunderstanding. The concept of masking in attention mechanisms was first introduced in the *"Attention Is All You Need"* paper [1], specifically for causal masking in auto-regressive tasks. The authors state: *"We need to prevent leftward information flow in the decoder to preserve the auto-regressive property. We implement this inside of scaled dot-product attention by masking out (setting to −∞) all values in the input of the softmax which correspond to illegal connections."* While this paper introduced masking for causal dependencies, the concept has since been extended to handle masked tokens in various contexts, such as masked language modeling in BERT [2]. Here is a more elaborate explanation of the masking approach.
>
> >$ \text{Softmax}(\frac{Q.K^T}{\sqrt{d}} + \eta M) $
>
> >Here, $\eta$ is a large negative value tending to −∞ (implemented as float(“-inf”) in PyTorch). M is a hard binary mask with values either 0 (for unmasked positions) or 1 (for masked positions). The $\eta M$ thus converts the attention scores for the masked positions (M = 1) into $\eta$, while leaving the unmasked scores unchanged.  Applying the softmax ensures $e^\eta = 0$, thus effectively removing the masked positions from the attention scores and preserving the relative probabilities of the unmasked tokens. Note that the bias term is **not a hyper-parameter** that requires to be fine-tuned, rather is fixed to a constant value of float(“-inf”).
>
> >An alternative to the above formulation could be applying M as a hard-constraint post-softmax as follows,
>
> >$ \text{Softmax}(\frac{Q.K^T}{\sqrt{d}}) \cdot (1 - M) $
>
> >Here, we first compute the normalized attention scores and then apply the mask by multiplying 1 – M to zero-out the masked positions. A drawback of this method would be that it would break the *sum of probability = 1* property across the final attention scores, leading to an invalid distribution.
>
> We are happy to clarify if there are any further questions about our formulation.
>
> >[1] Vaswani, A. "Attention is all you need." Advances in Neural Information Processing Systems (2017).
> [2] Devlin, Jacob, Ming-Wei Chang, Kenton Lee, and Kristina Toutanova. "BERT: Pre-training of Deep Bidirectional Transformers for Language Understanding." arXiv preprint arXiv:1810.04805 (2019)

---

> > ### Comment · Reviewer_UWVS · 2024-11-27
> > **Response to the authors' rebuttal (2nd round)**
> >
> > I thank the authors for the clarification. Here are my follow-up comments.
> >
> > 1. I thank the authors for additional experiments. While it's true that imputation methods prioritize interpolation over predictive power, the significant performance loss with SAITS remains concerning. The reported values deviate considerably from simple baselines, which raises questions about whether SAITS was properly trained with well-chosen hyperparameters.
> > Despite the new results and baselines, the authors haven't adequately explained why the proposed method excels only at longer time horizons. This behavior likely depends on the underlying temporal dynamics of the dataset. A more in-depth and systematic analysis, potentially incorporating varying temporal characteristics, is crucial to fully understand the advantages of the proposed method
> >
> > 2. While I was familiar with BERT and masking strategies in general, I hadn't previously encountered the use of mask indicators (with  $-\infty$) within the softmax function for attention score calculation as introduced in this paper. A more intuitive approach might involve a hard constraint (as mentioned in the rebuttal) that only includes non-masked elements in the softmax calculation. This would ensure the attention scores sum to one, similar to the approach used in causal self-attention for decoder structures. This will provide a clearer explanation of how unmeasured observations are excluded during attention.

---

> ### Author Response · Authors · 2024-11-28
>
> Thank you for the further intriguing questions. Here are our responses to these questions.
>
> **Comment 1: While it's true that imputation methods prioritize interpolation over predictive power, the significant performance loss with SAITS remains concerning. The reported values deviate considerably from simple baselines, which raises questions about whether SAITS was properly trained with well-chosen hyperparameters.**
> > For our experiments, we used the same default hyperparameters as specified in the original implementation of SAITS. We would like to emphasize that the poor performance of SAITS is a reflection on SAITS’s usability on ETTm2 and not our proposed framework (MissTSM does not use SAITS or any other imputation method inside its framework).
>
> **Comment 2: The authors haven't adequately explained why the proposed method excels only at longer time horizons. This behavior likely depends on the underlying temporal dynamics of the dataset. A more in-depth and systematic analysis, potentially incorporating varying temporal characteristics, is crucial to fully understand the advantages of the proposed method**
> > Thank you for this suggestion. To explain why MissTSM excels at longer time horizons, note that missing values tend to have more catastrophic effects on predictions made far out into the future, because of the compounding effects of missing information across time. Additionally, imputation methods primarily focus on capturing *local* trends to fill missing values and hence are less effective at longer horizons. Please note that the strengths of MissTSM are not limited to longer forecasting horizons alone; we find that MissTSM is useful in other challenging scenarios too. For example, on classification datasets, where the notion of forecasting horizons does not apply, our approach demonstrates competitive performance especially with large masking fractions. The real-world applicability of MissTSM is reflected in the experiments on the PhysioNet dataset, where we see a significant jump in MissTSM’s performance compared to imputation baselines.
>
> > Regarding the depth of our experimental analyses, we would like to note that we have conducted experiments over 183 variations of benchmark datasets with varying patterns of missing values and missing value fractions. This scale of experiments is unprecedented for any previous work in time-series modeling that handles missing values. Hence, while we agree that there is potential to further expand our analyses to more datasets, we believe that our current results comprehensively show the usefulness of MissTSM over a wide range of dataset variations. We will be happy to include additional experiments on any other time-series dataset that we have not included so far in the discussion period.
>
> **Comment 3: While I was familiar with BERT and masking strategies in general, I hadn't previously encountered the use of mask indicators (with −∞) within the softmax function for attention score calculation as introduced in this paper. A more intuitive approach might involve a hard constraint (as mentioned in the rebuttal) that only includes non-masked elements in the softmax calculation. This would ensure the attention scores sum to one, similar to the approach used in causal self-attention for decoder structures. This will provide a clearer explanation of how unmeasured observations are excluded during attention**
> > Thank you for suggesting this alternate formulation. Based on our understanding of the reviewer’s comment, this alternate formulation that includes hard constraints on masked tokens but inside the softmax computation can be expressed as:
> > $ \text{Softmax} (\frac{Q.K^T}{\sqrt{d}} \cdot (1 - M))$,
> where the inputs to the softmax are set to 0 if the mask M is 1. While this would ensure that the outputs sum to one, a limitation of this framework would be that the masked tokens would receive non-zero attention scores. In particular, softmax would transform the zero inputs of masked tokens to output $\frac{1}{N + \sum e^{U_i}}$, where $U_i$ represents unmasked tokens and $N$ is the number of masked tokens. Additionally, the outputs of the unmasked tokens would also be influenced by the number of masked tokens $N$ as follows: $\frac{e^{U_i}}{N + \sum e^{U_j}}$. To avoid these limitations, we set the inputs to the softmax to $–\infty$ if the mask M is 1. Note that this is an implementation detail that is often not fully described in the main paper but is included in code implementations of seminal works in Transformers, e.g., attention implementation of transformer model in PyTorch - https://pytorch.org/docs/stable/generated/torch.nn.functional.scaled_dot_product_attention.html#torch.nn.functional.scaled_dot_product_attention, attention layer in BERT - lines 705-716 in https://github.com/google-research/bert/blob/master/modeling.py. Our formulation of masked attention using $-\infty$ is thus not a novel contribution but a standard practice in the community

---

### Official Review · Reviewer_sZVS · 2024-11-13

**Soundness:** 2
**Presentation:** 3
**Contribution:** 2
**Rating:** 5
**Confidence:** 3

**Summary:**

The paper presents an imputation-free approach called MissTSM to handle missing data in time series. The MissTSM framework is innovative, and the experiments demonstrate its effectiveness.

**Strengths:**

1. The paper is well organized and clearly articulates the methodology behind MissTSM;
2. The authors provided a link to their source code for reproducibility;

**Weaknesses:**

1. Three datasets for each task are limited and not enough to verify a framework;
2. Too many experiment setup details are missing. For example, a). hardware information is missing in the computational cost comparison; b). the authors do not mention how they determine models' hyperparameters;

**Questions:**

1. How did the hyperparameters of the baseline models get optimized? Please discuss and reveal more details regarding this point.
2. Baseline models are missing in the code repository. Did the authors reproduce the baseline methods by using their official code or other unified Python libraries (e.g.  Time-Series-Library [1], PyPOTS [2])? Data processing is quite different across imputation algorithms while unified interfaces can ensure fairness in the experiments.
3. Now that only three datasets are for the forecasting task, why select both ETTh2 and ETTm2 from ETT datasets for experiments? Generally, datasets from different application domains are more reasonable.

### References
[1] https://github.com/thuml/Time-Series-Library

[2] Wenjie Du. PyPOTS: a Python toolbox for data mining on Partially-Observed Time Series. In KDD MiLeTS Workshop, 2023. https://github.com/WenjieDu/PyPOTS

---

> ### Author Response · Authors · 2024-11-24
>
> **Comment 1: Three datasets for each task are limited and not enough. Why both ETTh2 and ETTm2 are selected from ETT datasets for experiments? Generally, datasets from different application domains are more reasonable.**
> > We chose ETT datasets because they are commonly used in most previous works in time-series forecasting, such as [1, 2, 3]. While we agree that using three datasets for forecasting and three for classification appear small, please note that for every dataset, we have considered extensive variations of the same dataset (40x for forecasting and 20x for classification) to study the effects of missing values on time-series models (namely, with different types of synthetic masking schemes, varying masking fractions, and multiple random runs). This is a unique feature of our experiments that differentiates us from prior works in the literature that only consider one setting of every dataset. Since every variation of a benchmark dataset can be viewed as a different sample dataset for the purpose of evaluation, we are able to add sufficient diversity in the nature and scale of dataset variations (183 in total) considered in our experiments.
>
> > To further increase the diversity of datasets used for forecasting, we have added results on a new dataset – Solar-Energy [4] with MCAR masking scheme, summarized in the Table below. Here, we compare the MSE performance of MissTSM with iTransformer coupled with the SAITS-imputation method. The standard deviations are reported based on 3 runs for iTransformer.
> | Masking (%) | Horizon | iTransformer | MissTSM |
> |-------------|---------|--------------------------|---------------|
> | **60%**     | 96      | 0.213 ± 0.02            | 0.222         |
> |             | 192     | 0.234 ± 0.03            | 0.253         |
> |             | 336     | 0.237 ± 0.03            | 0.237         |
> |             | 720     | 0.233 ± 0.08            | 0.236         |
> | **70%**     | 96      | 0.232 ± 0.03            | 0.217         |
> |             | 192     | 0.255 ± 0.04            | 0.218         |
> |             | 336     | 0.268 ± 0.04            | 0.222         |
> |             | 720     | 0.254 ± 0.04            | 0.252         |
> | **80%**     | 96      | 0.215 ± 0.07            | 0.251         |
> |             | 192     | 0.237 ± 0.03            | 0.289         |
> |             | 336     | 0.248 ± 0.02            | 0.265         |
> |             | 720     | 0.240 ± 0.07            | 0.267         |
> | **90%**     | 96      | 0.264 ± 0.03            | 0.290         |
> |             | 192     | 0.298 ± 0.08            | 0.370         |
> |             | 336     | 0.305 ± 0.02            | 0.319         |
> |             | 720     | 0.296 ± 0.01            | 0.321         |
>
> >  We observe that MissTSM shows *competitive performance* as iTransformer across all masking fractions of Solar, similar to the trends observed on other forecasting datasets. This when coupled with the fact that our method is imputation-free makes it an ideal choice for handling missing values in time-series forecasting problems, because (a) it does not suffer from the computational costs of training complex imputation models such as SAITS, and (b) it does not suffer from the propagation of imputation errors to forecasting errors as is the case for other baselines, as described in Appendix C.
>
> >[1] Wu, Haixu, Jiehui Xu, Jianmin Wang, and Mingsheng Long. "Autoformer: Decomposition transformers with auto-correlation for long-term series forecasting." Advances in Neural Information Processing Systems 34 (NeurIPS) 2021
> [2] Liu, Yong, Tengge Hu, Haoran Zhang, Haixu Wu, Shiyu Wang, Lintao Ma, and Mingsheng Long. "iTransformer: Inverted transformers are effective for time series forecasting.” International Conference on Learning Representations, (ICLR) 2024
> [3] Dong, Jiaxiang, Haixu Wu, Haoran Zhang, Li Zhang, Jianmin Wang, and Mingsheng Long. "Simmtm: A simple pre-training framework for masked time-series modeling." Advances in Neural Information Processing Systems 36 (NeurIPS) 2023
> [4] Lai, Guokun, Wei-Cheng Chang, Yiming Yang, and Hanxiao Liu. "Modeling long-and short-term temporal patterns with deep neural networks." In The 41st international ACM SIGIR conference on research & development in information retrieval, pp. 95-104. 2018.

---

> > ### Author Response · Authors · 2024-11-24
> >
> > **Comment 2: Too many experiment setup details are missing. For example, a) hardware information is missing; b) the authors do not mention how they determine the hyper-parameters of their model and baselines; c) Baseline models are missing in the code repository, missing details of their implementation.**
> >
> > > We apologize for missing these details in the original submission, which we are happy to include as follows.
> >
> > >1. We performed our computational complexity experiments on NVIDIA 24G GPUs.
> > >2. We use default hyper-parameters as reported in the official codebases of the baseline models. For MissTSM, we start with the same initial set of hyper-parameters as reported in the SimMTM paper and then search for the best learning rate in factors of 10, and the number of encoder/decoder layers in the range of 2 to 4. Our motivation behind this simple scheme of selecting hyper-parameters to show the generic effectiveness of MissTSM framework even without rigorous hyper-parameter optimization.
> > >3. The baseline models are implemented using their official codebases except for SAITS, for which we used the PyPOTS library.

---

> > > ### Author Response · Authors · 2024-11-30
> > >
> > > Thank you for your review. As we approach the end of the discussion period, we would like to extend an invitation for any further questions or clarifications. If our response has been adequate in addressing your concerns, we kindly request you to consider revising your scores accordingly. Thank you very much for your valuable feedback.

---

### Author Response · Authors · 2024-11-24
**Global Response to Review Comments**

We sincerely thank all the reviewers for providing constructive feedback. We are encouraged that the reviewers found
- Our proposed idea to be intuitive and clearly motivated [*Reviewer 7Jfe*]
- Our evaluation to be comprehensive with multiple imputation baselines and time-series tasks, including forecasting and classification [*Reviewers G7Mn, 7Jfe*]
- Our paper to be well-written and easy to follow [*Reviewers sZVS, UWVS, 7Jfe*]

Before addressing each of the reviewer’s comments individually, we address some of the shared concerns raised by the reviewers in our general response below:

**Comment 1: Improvement in MissTSM’s performance appears marginal (UWVS, 7Jfe)**
> We acknowledge that MissTSM’s performance improvement over other baselines may appear marginal in some scenarios and is not always the best-performing model across the extensive set of experiments considered in this paper, which is difficult to achieve given the natural variations expected in most empirical analyses. However, MissTSM shows *competitive performance* close to the best-performing model *consistently* across varying datasets, horizon windows, masking schemes, and masking fractions.

> To quantify this claim, we have now added a new column in Table 1 that shows the average rank for every method across all three forecasting datasets and horizon windows for every synthetic masking scheme (using a fixed masking fraction). From Table 1, we can see that MissTSM is a *close second* in all three masking schemes with average ranks of 1.9, 2.7, and 4.1 for no masking, MCAR masking, and periodic masking, respectively. However, the average ranks of Table 1 still do not comprehensively summarize performance trends across all experiments of the paper. Instead, the performance of MissTSM varies across datasets (we are consistently better than other baselines on ETTh2 but not on Weather), horizon windows, and masking fractions, as presented in Figures 4 to 7 of the main paper. We make the following **key remarks** summarizing the comparison of MissTSM with baselines across all experiments of our paper:
> 1. MissTSM shows significant performance gains for larger masking fractions such as 90% (see Figure 4) and larger horizon windows such as 720 (see Figure 5). This suggests that MissTSM is particularly useful in extreme testing environments where other baseline methods break down.
2. MissTSM shows worse performance than baselines only in two scenarios: EMG dataset (classification) with masking fraction 0.2, and Weather dataset (forecasting) for smaller horizon windows (less than 720). This suggests that there is scope for improving MissTSM for smaller masking fractions and horizon windows.
3. While the 6 benchmark datasets used in our work are currently the de facto standards in the time-series modeling community, these datasets also have their weaknesses. For example, we can see from Table 1 that the spline interpolation generally performs better if not at par with the more complex SAITS imputation, suggesting that these datasets are simpler with regular periodic structures than what we may expect in real-world settings.
4. On Physionet, a real-world dataset with 80% missing values, we can see from Figure 8 that MissTSM performs significantly better than all other baselines. This suggests the practical utility of our work on datasets with real patterns of missing values, as opposed to datasets generated from synthetic masking schemes that may have limitations.

> Apart from these four remarks on the performance of MissTSM, we would also like to emphasize that MissTSM is imputation-free and hence (a) does not suffer from the computational costs of imputation, and (b) does not involve error propagation from imputation to forecasting (see Appendix C), making it an ideal choice for handling missing values in multivariate time-series data.

**Comment 2: More baselines need to be included in the experiments (UWVS, G7Mn)**
> We have performed new experiments with GRU-D and Latent ODE methods for classification and forecasting tasks, as suggested by [UWVS] and with RobustTSF as suggested by [G7Mn]. Our results indicate that MissTSM significantly outperforms these baselines in their respective tasks and datasets.

We hope that our responses to the individual review comments provided below address the main concerns of the reviewers. If we missed any detail, we will be happy to provide more clarifications during the discussion period. If our responses have adequately addressed your concerns, we kindly request that you consider revising your scores. Thank you very much for your time and effort.

---

> ### Author Response · Authors · 2024-12-02
> **Closing Remarks and Request for Final Reviewer Comments and Score Updates**
>
> We sincerely thank the reviewers for their insightful comments and the constructive feedback, which has been immensely helpful in improving the quality of our work. As the discussion period draws to a close, we invite you to share any remaining questions or requests for clarifications. We take this opportunity to re-emphasize the main contributions of our work, and summarize the main concerns raised by the reviewers and our responses to those concerns below.
>
> **Main contributions:**
>
> (1) We introduce a time-series modeling framework that can handle missing values directly within the framework of Masked Autoencoders (MAE), thus bringing in the power of representation learning and masked pre-training to the domain of time-series modeling.
>
> (2) Our proposed framework is simple, fast, lightweight and scalable compared to the complex two-stage imputation-based approaches and existing time-series models for handling missing values, making it an effective choice for practitioners especially in resource-constrained settings.
>
> (3) We show that our proposed framework consistently achieves competitive performance as the SOTA models under diverse synthetically generated missing value scenarios on benchmark datasets for forecasting and classification, especially for larger masking fractions and longer horizon windows. Our framework also demonstrates superior performance on a real-world dataset for classification.
>
> **Main reviewer concerns and responses**
>
> **Concern 1: The work presents a limited perspective on time-series methods that avoid imputation. More baselines need to be included in the experiments (UWVS, G7Mn)**
>
> >**Response:** We added new experiments comparing our work with three more baselines as suggested by reviewers.
> >- We compared MissTSM with GRU-D on three classification datasets (see our response to *Reviewer UWVS*)
> >- We compared MissTSM with LatentODE on the ETTh2 forecasting dataset across 4 different missing value fractions (see our response to *Reviewer UWVS*)
> >- We compared MissTSM with RobustTSF on the ETTh2 forecasting dataset, across 4 different missing value fractions (see our response to *Reviewer G7Mn*)
>
> > In all the experiments, we see that MissTSM outperforms baselines by significant margins, further bolstering the contribution of this work in relation to previous works.
>
> **Concern 2: Improvement in MissTSM’s performance appears marginal (UWVS, 7Jfe)**
> >**Response:** As discussed in our global response, MissTSM consistently shows competitive performance across diverse datasets and shows significant performance gains in challenging scenarios such as high masking fractions and large horizon windows. MissTSM also significantly outperforms baselines on a real-world dataset, Physionet, showcasing its practical applicability. Moreover, the imputation-free nature of our model prevents the computational overhead and propagation of imputation errors to forecasting errors.
>
> **Concern 3: Lack of variety in datasets (sZVS)**
>
> >**Response:** While our study was limited to only 6 datasets, we have conducted experiments with extensive variations of each dataset (40x for forecasting and 20x for classification), contrary to prior works in the literature. Overall, we have conducted experiments on over 183 variations of benchmark datasets with varying patterns of missing values and missing value fractions. This scale of experiments is unprecedented for any previous work in time-series modeling that handles missing values. To further increase the diversity of datasets, we added results on a new forecasting dataset, Solar-Energy (see our response to *Reviewer  sZVS*) and observe that MissTSM shows similar trends in performance as observed on other forecasting datasets.
>
> **Concern 4: The experiments consider only high missing rates 60-90% (G7Mn, 7Jfe)**
>
> >**Response:** We considered moderate to extreme ranges of missing value fractions in our experiments to closely resemble real-world scenarios where the missing value levels are very high. To show results on smaller missing value fractions, we conducted additional experiments on ETTh2 dataset across 10-40% missing data fractions (see our response to *Reviewer G7Mn*). Through these results, we see that while MissTSM is very effective under challenging conditions it also gives good performance under simpler scenarios with low missing data.

---

> > ### Author Response · Authors · 2024-12-02
> > **[Continued] Closing Remarks and Request for Final Reviewer Comments and Score Updates**
> >
> > **Concern 5: Discuss how the current approach might be extended or adapted to handle other types of data quality problems (G7Mn)**
> >
> > >**Response:** While the original focus of this work was on handling missing values, we conducted new experiments to show the applicability of MissTSM for handling anomalies based on reviewer suggestions (see our response to *Reviewer G7Mn*). Through these experiments, we see that simple extensions of MissTSM can be easily developed to handle other data quality issues such as anomalies and show comparable or even better performance than state-of-the-art baseline methods such as RobustTSF. This is a testament to the broader applicability of our work outside missing values, motivating other extensions of MissTSM in future works.
> >
> >
> > We hope the above summary comprehensively covers the main topics of the discussion period; we apologize if we missed any detail. If our responses have adequately addressed your concerns, we kindly request that you consider revising your scores. Thank you very much for your time and effort.

---

### Meta-Review · Area_Chair_V3vk · 2024-12-20

**Metareview:**

This paper proposes an imputation-free approach called MissTSM for handling missing values in time series data, using Time-Feature Independent embedding and Missing Feature-Aware Attention mechanisms. The paper has a clear presentation and comprehensive experimentation. However, as raised by the reviewer, the novelty and the experiments are somewhat limited.

**Additional Comments On Reviewer Discussion:**

Although the paper has some merits, such as clear presentation and comprehensive experimentation, the issues raised by the reviewers are critical. For instance, the limited novelty of the technical contributions and marginal performance improvements (UWVS), the insufficient coverage of related imputation-free approaches and missing state-of-the-art baselines (G7Mn), and the limited experimental setup (sZVS). Although the authors include some additional experiments in their response, the paper still needs a major revision before it can be accepted, especially in addressing the fundamental concerns about technical novelty.

---

### Decision · Program_Chairs · 2025-01-22

Reject